# Activation Sharing with Asymmetric Paths Solves Weight Transport Problem without Bidirectional Connection

**Sunghyeon Woo**   **Jeongwoo Park**   **Jiwoo Hong**   **Dongsuk Jeon**

Seoul National University

{wsh0917, jeffjw, micro995, djeon1}@snu.ac.kr

## Abstract

One of the reasons why it is difficult for the brain to perform backpropagation (BP) is the weight transport problem, which argues forward and feedback neurons cannot share the same synaptic weights during learning in biological neural networks. Recently proposed algorithms address the weight transport problem while providing good performance similar to BP in large-scale networks. However, they require bidirectional connections between the forward and feedback neurons to train their weights, which is observed to be rare in the biological brain. In this work, we propose an Activation Sharing algorithm that removes the need for bidirectional connections between the two types of neurons. In this algorithm, hidden layer outputs (activations) are shared across multiple layers during weight updates. By applying this learning rule to both forward and feedback networks, we solve the weight transport problem without the constraint of bidirectional connections, also achieving good performance even on deep convolutional neural networks for various datasets. In addition, our algorithm could significantly reduce memory access overhead when implemented in hardware.

## 1 Introdution

Backpropagation (BP) [1] is the representative approach to training various deep neural networks. While BP exhibits excellent training performance, similar to or even better than that of humans, it has been long argued that the structure of biological neural networks does not support the backpropagation of errors [2, 3, 4, 5]. To resolve this issue, a wide range of studies have been conducted to develop an algorithm that is feasible in biological neural networks. One important reason behind the biological implausibility of BP is the *weight transport problem* [6]. BP requires identical forward and feedback paths for reliable training; i.e., the two paths must have the same synaptic weights. While biological neural networks may also implement two separate processing paths (forward and feedback paths), it is impossible to explicitly pass the weights between the two paths, as it requires a very fast transmission of information along the axon from each synapse output [7].

The Feedback Alignment (FA) algorithm [8] solves the weight transport problem by propagating errors through the feedback path with random fixed weights. It was shown that the feedback weights could be aligned with the forward weights in the course of training, and consequently, the network is trained in a similar way to BP. Similarly, Direct Feedback Alignment (DFA) [9] directly propagates errors from the top layer to lower layers using random fixed weights, and Direct Random Target Projection [10] locally creates errors using targets rather than using backpropagated errors. Although these algorithms demonstrate good training performance on simple networks, they exhibit large performance degradation when applied to complex networks, especially deep convolutional neural networks [11, 12].

35th Conference on Neural Information Processing Systems (NeurIPS 2021).

More recent works suggest training the feedback weights as well as the forward weights instead of using random fixed weights in the feedback path. Training feedback weights through Reinforcement Learning [13] or Spiking Neural Network [14] provides better accuracy in classification tasks than the algorithms using random fixed weights. However, those approaches have been only evaluated in shallow neural networks, and it is questionable whether they would be capable of training deeper neural networks. The Sign-Symmetry algorithm [15] achieves good training performance in deep neural networks by transporting only the signs of the forward weights to the feedback path during training. However, since it still needs to transport the signs of the weights, it is difficult to say that the weight transport problem has been solved entirely.

Recently, Weight Mirrors and modified Kolen-Pollack algorithms [16, 17], which align the feedback weights with the forward weights without explicit weight transport, succeeded in training deep convolutional neural networks on large datasets with outstanding performance close to BP. However, these algorithms require a bidirectional connection: forward and feedback neurons explicitly exchange information with each other through a pair of direct unidirectional paths. While this type of connection is found in some biological neurons [18, 19, 20, 21], it is difficult to generalize them to a broader range of biological neural networks that typically only have unidirectional paths between neurons, where feedback must route through multiple neurons in order to reach their destination [22].

In this paper, we propose an Activation Sharing algorithm, which performs weight update of a layer using the output (*activation*) of a lower layer, not the output of the immediately previous layer. Applying this learning method to both forward and feedback paths solves the weight transport problem without using bidirectional paths and also successfully trains deep convolutional networks such as ResNet-34. Furthermore, since activation is shared across multiple layers during training, the algorithm could significantly reduce memory access and is suitable for efficient hardware implementation.

## 2  Related works

### 2.1  Backpropagation and weight transport problem

We consider typical training process of a deep neural network. The forward path propagates input data through multiple layers, generating output activation of each layer:

$$\boldsymbol{h}_{l+1} = \phi(\boldsymbol{W}_{l+1}\,\boldsymbol{h}_l + \boldsymbol{b}_{l+1}) \tag{1}$$

where $\boldsymbol{h}_l$ is the output activation of the hidden layer $l$, $\boldsymbol{W}_l$ is the weight matrix, and $\boldsymbol{b}_l$ is the bias vector of layer $l$ in the forward path. In the brain, $\boldsymbol{h}$ could be interpreted as neural firing rates, $\boldsymbol{W}$ as synaptic weights between neurons, $\boldsymbol{b}$ as bias currents, and $\phi$ as nonlinearities in neurons [16]. To train the model, synaptic weight changes $\Delta \boldsymbol{W}$ are obtained by

$$\Delta \boldsymbol{W}_{l+1} = -\eta\,\boldsymbol{\delta}_{l+1}\,\boldsymbol{h}_l^T \tag{2}$$

where $\boldsymbol{\delta}$ and $\eta$ denote the error signal that flows through the feedback path and the learning rate, respectively. For calculating synaptic weight changes $\Delta \boldsymbol{W}$, the error $\boldsymbol{\delta}$ should propagate from the output layer to lower layers in the reverse direction. In backpropagation [1], errors that propagate through the feedback path are represented as

$$\boldsymbol{\delta}_l = \phi^{'}(\boldsymbol{h}_l)\,\boldsymbol{W}_{l+1}^T\,\boldsymbol{\delta}_{l+1} \tag{3}$$

where $\phi^{'}$ denotes the derivative of the activation function $\phi$ in equation (1). For this BP to occur in the brain, the error must be propagated in the reverse direction of the forward path, which is not feasible in biological neural networks that only have unidirectional paths. Even if there is a separate feedback path, as shown in Fig. 1a, the synaptic weights $\boldsymbol{W}$ in equation (1) should be also used to propagate the error in equation (3). In other words, the forward and feedback paths must have identical synaptic weights. Neurons need to exchange their synaptic weights to realize this, but there is no proof that this weight transport actually occurs in the brain (*weight transport problem*).

### 2.2  Training with random fixed feedback weights

The Feedback Alignment algorithm [8] was developed based on the idea that errors could be propagated using random fixed weights $\boldsymbol{B}$ rather than the forward weights $\boldsymbol{W}$ in equation (1). As shown in Fig. 1b, errors propagate through the feedback path following the equation below.

$$\boldsymbol{\delta}_l = \phi^{'}(\boldsymbol{h}_l)\,\boldsymbol{B}_{l+1}^T\,\boldsymbol{\delta}_{l+1} \tag{4}$$

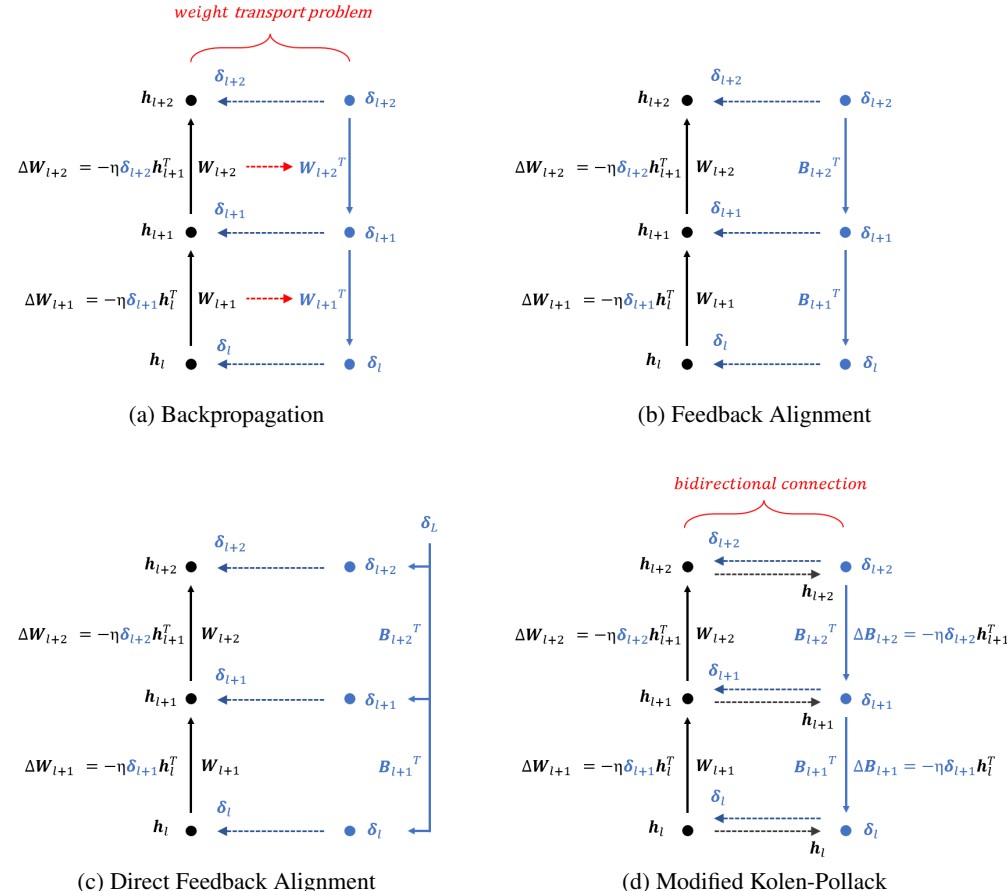

(a) Backpropagation

(b) Feedback Alignment

(c) Direct Feedback Alignment

(d) Modified Kolen-Pollack

Figure 1: Backpropagation and biologically plausible algorithms.

Moreover, the Direct Feedback Alignment algorithm [9] showed that the neural network could be trained if the error propagates directly from the output layer to each layer instead of flowing through all hidden layers sequentially (Fig. 1c):

$$\boldsymbol{\delta}_l = \phi^{'}\left(\boldsymbol{h}_l\right) \boldsymbol{B}_{l+1}^T \boldsymbol{\delta}_L \tag{5}$$

where $L$ denotes the output layer of a deep neural network. These methods perfectly solve the weight transport problem by using random fixed weights in the feedback path and demonstrate good training performance in simple networks. However, they fail to train the model or incur significant performance drop when applied to deep convolutional neural networks [11, 12, 15, 23].

## 2.3 Training with feedback weight update

Since using random fixed weights for error propagation causes poor performance for training deep and complex neural networks, various methods that train the feedback weights simultaneously have been recently proposed [13, 14, 24]. For example, the Weight Mirror (WM) and modified Kolen-Pollack (KP) algorithms attain very competitive training performance on deep convolutional networks [16]. In WM, the feedback weights are updated by

$$\Delta \boldsymbol{B}_{l+1} = -\eta \, \boldsymbol{h}_{l+1} \, \boldsymbol{h}_l^T \tag{6}$$

using output activations of layers $l$ and $l+1$ while the error propagates in the same way as FA according to equation (4). This update rule makes $\Delta \boldsymbol{B}_{l+1}$ proportional to $W_{l+1}$, which aligns the feedback weights with the forward weights during training as discussed in [16]. On the other hand, the KP algorithm updates the feedback weights as below:

$$\Delta \boldsymbol{B}_{l+1} = \Delta \boldsymbol{W}_{l+1} = -\eta \, \boldsymbol{\delta}_{l+1} \, \boldsymbol{h}_l^T \tag{7}$$

This method supports updating forward and feedback weights in the same direction, and those two weights could also be aligned during training. These algorithms succeed in training deep convolutional neural networks with similar performance to BP. Both WM and KP solve the weight transport problem by avoiding explicitly exchanging information related to synaptic weights such as weight changes $\Delta W$ and $\Delta B$ between the two paths; instead, a forward neuron and its corresponding neuron in the feedback path exchange activation $h$ and error $\delta$. For example, $\delta_{l+1}$ of the feedback path is needed to update the forward weights $W_{l+1}$ in equation (2), while $h_{l+1}$ of the forward path is required to update the feedback weights $B_{l+1}$ in equation (6) in WM. KP also needs $h_{l+1}$ to update the feedback weights $B_{l+2}$ in equation (7). However, exchanging information between two neurons constitutes a bidirectional connection, which is not supported in general biological neural network structures consisting of only unidirectional connections.

## 3 Activation Sharing to solve weight transport problem without bidirectional connections

Prior studies to find biologically plausible algorithms have mainly focused on approximating the forward path without modifying the forward path for training. However, the demand for correct activations to update weights inevitably incurs bidirectional connections when training feedback weights as depicted in Fig. 1d. Our study, on the other hand, develops a learning algorithm not relying on accurate activations. In this section, we first review the Feedback Alignment algorithm, which uses an approximate feedback path. Based on this, we show that the forward path could also be approximated, and we present an Activation Sharing algorithm. Finally, we show that the algorithm solves the weight transport problem without bidirectional connections in the network.

### 3.1 Approximate feedback path: Feedback Alignment

For the sake of simplicity, here we consider a linear network (i.e., we assume $h_{l+1} = W_{l+1}\,h_l$ neglecting $\phi$). As shown in equation (2), we need the error $\delta$ and the activation $h$ to update the weights in the network. For backpropagation, we could calculate these values very accurately. We use the activations obtained when the input propagates through layers and the errors that back-propagate through the same layers in the reverse order. Since we assume a linear network, equation (3) is simplified to $\delta_{l+1} = W_{l+2}^T\,\delta_{l+2}$. Consequently, the learning process in backpropagation (Fig. 2a) follows the equations below.

$$h_{l+1} = W_{l+1}\,h_l \tag{8}$$

$$\delta_{l+2} = W_{l+3}^T\,\delta_{l+3} \tag{9}$$

$$\Delta W_{l+2} = -\eta\,\delta_{l+2}\,h_{l+1}^T \tag{10}$$

However, recent studies suggest that approximate errors could also be used for weight updates. For instance, Feedback Alignment uses approximate errors $\tilde{\delta}$ that propagate through random fixed weights. The learning process is described in Fig. 2b and is expressed by

$$h_{l+1} = W_{l+1}\,h_l \tag{11}$$

$$\tilde{\delta}_{l+2} = B_{l+3}^T\,\tilde{\delta}_{l+3} \neq \delta_{l+2} \tag{12}$$

$$\Delta W_{l+2} = -\eta\,\tilde{\delta}_{l+2}\,h_{l+1}^T \neq -\eta\,\delta_{l+2}\,h_{l+1}^T \tag{13}$$

where $\tilde{\delta}_L = \delta_L$ since the error at the output layer due to the loss function is the same for BP and FA. This approximate feedback path successfully trains a neural network by aligning the forward weights $W$ with the random fixed weights $B$.

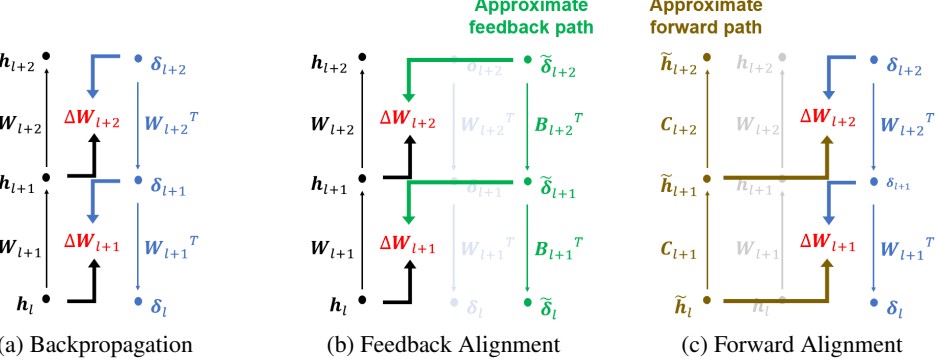

(a) Backpropagation     (b) Feedback Alignment     (c) Forward Alignment

Figure 2: Overview of weight update process in different algorithms.

## 3.2    Approximate forward path: Forward Alignment & Activation Sharing

The Feedback Alignment algorithm showed that approximate errors could be used to train the network. Similarly, we hypothesize that the network could be trained even if we use approximate activations $\tilde{h}$ obtained by propagating the input through random fixed weights $C$ instead of the exact forward path, which we dub *Forward Alignment*. The flow of the Forward Alignment is described in Fig. 2c and is expressed by the equations below.

$$\tilde{h}_{l+1} = C_{l+1}\, \tilde{h}_l \neq h_{l+1} \tag{14}$$

$$\delta_{l+2} = W_{l+3}^T\, \delta_{l+3} \tag{15}$$

$$\Delta W_{l+2} = -\eta\, \delta_{l+2}\, \tilde{h}_{l+1}^T \neq -\eta\, \delta_{l+2}\, h_{l+1}^T \tag{16}$$

where $\tilde{h}_0 = h_0$ since the network input is identical in both cases. The Feedback Alignment algorithm was proven to work in simple linear networks through Barbalat's lemma that the loss of the network converges to zero if certain conditions are satisfied [25]. Likewise, it can be shown that the loss converges to zero by Forward Alignment in simple linear networks through Barbalat's lemma. Experimental results confirm that the Forward Alignment algorithm aligns the forward weights with the random fixed weights during training (see Appendix A for details).

Extending this idea, we propose an Activation Sharing algorithm that updates the weights using the activations of a lower layer based on two intuitions. First, since the Forward Alignment algorithm works with any random fixed weights, the identity matrix $I$ could also be used for generating approximate activations when the adjacent layers have the same input dimension. Second, propagating the input $h_0$ through more layers with random fixed weights to obtain approximate activations $\tilde{h}$ would incur more deviations from the forward activations $h$ since $\tilde{h}_l = C_l\, \tilde{h}_{l-1} = \prod C_k\, h_0$. On the other hand, if we limit the propagation of input or activation through random fixed weights for approximate activation generation within a *block*, which is defined as a set of multiple consecutive layers in the network, this deviation would be significantly reduced, and the training process would more closely follow the backpropagation. For instance, if each block consists of two layers, $\tilde{h}_l = C_l\, C_{l-1}\, h_{l-2}$, which would be closer to $h_l$ than $\tilde{h}_l = \prod C_k\, h_0$. Combining these two intuitions, we propose the Activation Sharing algorithm with a block size of 2 as follows:

$$\tilde{h}_{l+1} = C_{l+1}\, \tilde{h}_l = h_l \tag{17}$$

$$\tilde{h}_{l+2} = C_{l+2}\, \tilde{h}_{l+1} = C_{l+2}\, C_{l+1}\, \tilde{h}_l = h_l \tag{18}$$

$$\Delta W_{l+2} = -\eta\, \delta_{l+2}\, \tilde{h}_{l+1}^T = -\eta\, \delta_{l+2}\, h_l^T \tag{19}$$

$$\Delta W_{l+3} = -\eta\, \delta_{l+3}\, \tilde{h}_{l+2}^T = -\eta\, \delta_{l+3}\, h_l^T \tag{20}$$

By equations (19) and (20), the algorithm updates the weights of all the layers inside a block using the same activation $\boldsymbol{h}_l$, which is the output activation of the previous block (equations (17) and (18)). In other words, once the activations to be shared are determined, the weights are updated using the designated activations instead of their own activations for $k$ layers when the block size is $k$. Fig. 3a depicts the Activation Sharing algorithm when $k = 2$. If $k = L - 1$, all layers except the first layer share the output activations of the first layer to update the weights.

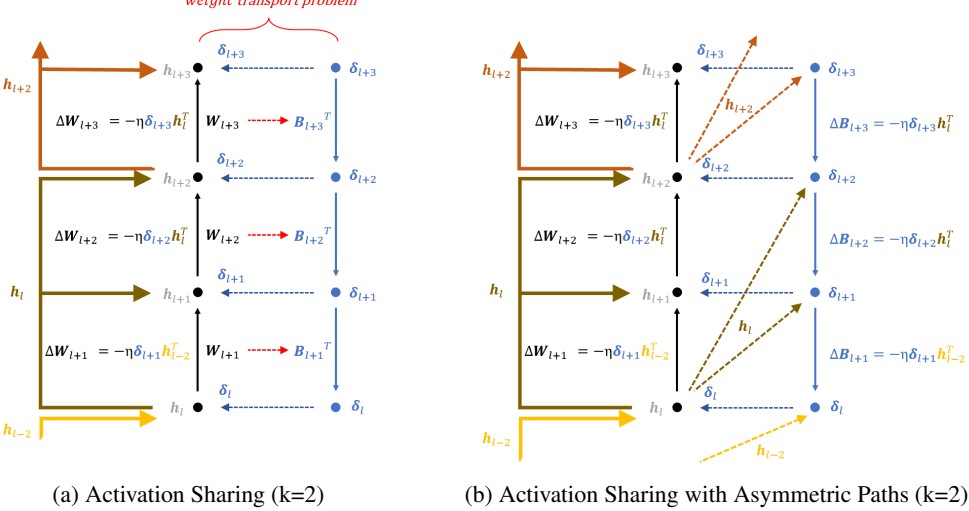

(a) Activation Sharing (k=2)       (b) Activation Sharing with Asymmetric Paths (k=2)

Figure 3: Activation Sharing & Activation Sharing with Asymmetric Paths.

### 3.3 How does Activation Sharing solve weight transport problem without bidirectional connections?

Kolen and Pollack [17] showed that when $\Delta \boldsymbol{W} = \Delta \boldsymbol{B}$ where $\boldsymbol{W}$ and $\boldsymbol{B}$ are the forward and feedback weights, the network is trained similarly to a typical artificial neural network with symmetric forward and feedback paths ($\boldsymbol{W} = \boldsymbol{B}$). For proof, the weight changes with weight decay [26] are written as

$$\Delta \boldsymbol{W}(t) = \boldsymbol{R}(t) - \lambda \boldsymbol{W}(t) \tag{21}$$

$$\Delta \boldsymbol{B}(t) = \boldsymbol{R}(t) - \lambda \boldsymbol{B}(t) \tag{22}$$

where $\boldsymbol{R}(t)$ denotes the weight change obtained by the learning rule in equation (2), and $\lambda$ denotes a weight decay factor. Then, $\boldsymbol{W}$ and $\boldsymbol{B}$ converge to identical values because $\boldsymbol{W}(t+1) - \boldsymbol{B}(t+1) = (\boldsymbol{W}(t) + \Delta \boldsymbol{W}(t)) - (\boldsymbol{B}(t) + \Delta \boldsymbol{B}(t)) = (1 - \lambda)(\boldsymbol{W}(t) - \boldsymbol{B}(t)) = (1 - \lambda)^{t+1}(\boldsymbol{W}(0) - \boldsymbol{B}(0))$ goes to 0 when $0 < \lambda < 1$ [16, 17].

This logic also applies to the Activation Sharing algorithm. As shown in Fig. 3a, the Activation Sharing algorithm itself does not solve the weight transport problem because it still assumes symmetric forward and feedback paths ($\boldsymbol{W} = \boldsymbol{B}$). However, the algorithm could be further modified to avoid the weight transport problem by relaxing this constraint and only enforcing $\Delta \boldsymbol{W} = \Delta \boldsymbol{B}$ in training. Specifically, we assume asymmetric forward and feedback paths (i.e., $\boldsymbol{W} \neq \boldsymbol{B}$) and apply the Activation Sharing algorithm to both paths using the same shared activations as shown in Fig. 3b. Since the forward and feedback neurons share the same activations, we can guarantee the following:

$$\Delta \boldsymbol{W}_{l+2} = \Delta \boldsymbol{B}_{l+2} = -\eta \, \boldsymbol{\delta}_{l+2} \, \boldsymbol{h}_l^T \tag{23}$$

$$\Delta \boldsymbol{W}_{l+3} = \Delta \boldsymbol{B}_{l+3} = -\eta \, \boldsymbol{\delta}_{l+3} \, \boldsymbol{h}_l^T \tag{24}$$

Therefore, the forward and feedback weights converge to the same values as discussed above, which enables the training of deep neural networks while solving the weight transport problem. Moreover, by relaxing the constraints that each layer must use its own input activations to update the weights,

bidirectional connections between forward and feedback neurons are not required anymore. We name this algorithm Activation Sharing with Asymmetric Paths (ASAP) to distinguish it from the Activation Sharing algorithm using symmetric paths discussed in Section 3.2. Code is available at: https://github.com/WooSunghyeon/Activation-Sharing-with-Asymmetric-Paths.

# 4 Biological implementation of ASAP algorithm

In previous sections, we propose the ASAP algorithm based on the observation that an approximate forward path allows for reliable neural network training. However, one question still remains: Is the ASAP algorithm actually biologically plausible? In other words, how is this algorithm implemented in biological neural networks? This section discusses the bio-plausibility of the ASAP algorithm in detail.

## 4.1 Relaxing the constraints of bidirectional connections

Some prior biologically plausible learning algorithms require bidirectional connections between forward and feedback neurons [16]. While one-to-one paring between the two neurons is observed in some organisms [18, 19, 20, 21], there exist structural constraints derived from bidirectional connections, as discussed in [18, 21]. Our ASAP algorithm could significantly relax these structural constraints by removing strict dependency between forward and feedback neurons depicted in Fig. 1d. ASAP also has dependency between forward and feedback neurons through multiple steps. However, this multi-step dependency is more strongly supported by biological observations. Reciprocal connections in the brain are formed not through direct bidirectional connections between two neurons, but through multiple intra-inter laminar routes using one-way connections [27]. For example, in the visual cortex, feedforward neurons of layer 4 cannot directly connect to feedback neurons of layer 3A, but only with the feedback neurons of layer 6 [27]. In addition, reciprocal connections are generally spatially asymmetrical. This is supported by the observation that feedforward connections form concentrated terminal arborizations, whereas feedback connections exhibit dispersed terminal arborizations [27, 28, 29]. Moreover, many unidirectional connections are observed in the feedback direction. In area TEO, neurons receive input from areas 35 and 36, but they do not project themselves into these perirhinal cortical regions [29, 30]. These observations show that our spatially asymmetric ASAP algorithm makes more sense than previous algorithms using simple and symmetrical bidirectional connections.

In summary, our ASAP algorithm mitigates the structural constraints of strict two-step dependencies by removing the need for direct information exchange between forward and feedback neurons. Furthermore, it can be implemented in biological neural networks using multiple one-way skip connections, which are connections between non-adjacent layers and are frequently found in living organisms [31, 32, 33, 34]. This suggests the ASAP algorithm is easier to implement biologically compared to other algorithms relying on bidirectional connections.

## 4.2 Learning with neuron-specific signals

The learning rules for biological neural networks (e.g., Hebbian learning) typically use neuron-specific signals; they use the information available in pre- or post-synaptic neurons for updating weights. In ASAP, however, some synapses require other information not directly available in pre- or post-synaptic neurons. For example, Fig. 3b shows that $\Delta W_{l+2}$ does not use pre-synaptic neuron activity $h_{l+1}$ but uses the neuronal activity of a lower layer $h_l$. The post-synaptic neurons may receive these activities as they do for error signals $\delta_{l+2}$ and use them for weight updates, but this necessitates numerous connections at the post-synaptic neuron since each synapse requires a synapse-specific signal (i.e., a different element of $h_l$). This type of synapse-specific signal transport is difficult to implement in biological neural networks. This issue could be mitigated by adopting the concept of the mirror mode in the Weight Mirror algorithm [16]. The Weight Mirror algorithm divides the learning process into two phases: engaged mode and mirror mode. In mirror mode, the feedback neuron mimics the forward neuron to generate the same activations.

Our ASAP algorithm could be realized using only neuron-specific signals in a similar way. In Fig. 4a, ASAP uses the error signals available in pre-synaptic neurons ($\delta_{l+2}$ and $\delta_{l+3}$) and the signals received by post-synaptic neurons ($h_l$) for updating feedback weights ($B_{l+2}$ and $B_{l+3}$). To update

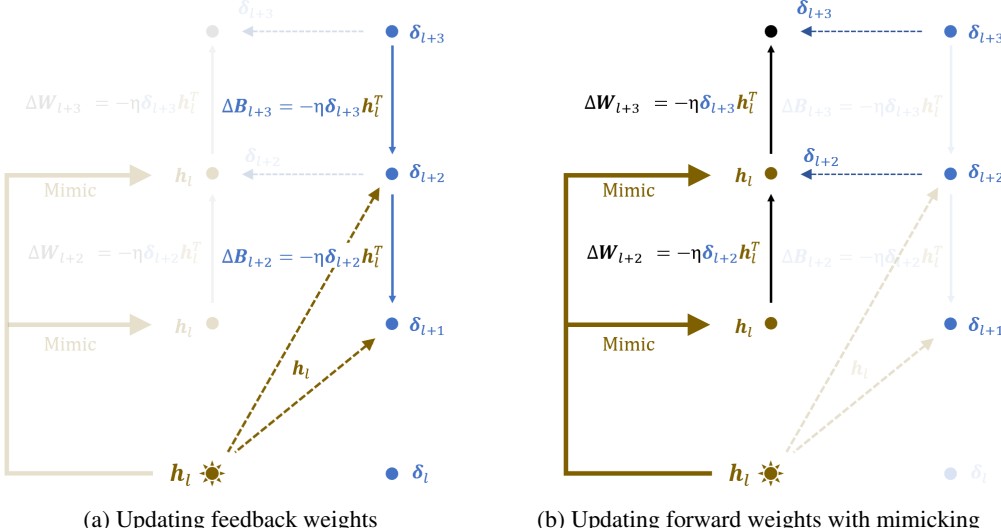

| (a) Updating feedback weights | (b) Updating forward weights with mimicking |

Figure 4: ASAP implementation using neuron-specific signals.

forward weights ($\boldsymbol{W}_{l+2}$ and $\boldsymbol{W}_{l+3}$), forward neurons mimic the shared activation ($\boldsymbol{h}_l$). Then, the pre-synaptic neuron activity ($\boldsymbol{h}_l$) and the signals received by post-synaptic neurons ($\boldsymbol{\delta}_{l+2}$ and $\boldsymbol{\delta}_{l+3}$) are used for learning as shown in Fig. 4b. Consequently, both forward and feedback weights are updated only using neuron-specific signals.

Furthermore, the forward and feedback weight updates can occur simultaneously in ASAP. In the Weight Mirror algorithm, the feedback neurons generate error signals $\boldsymbol{\delta}_{l+1}$, and the forward neurons receive them to update the forward weights $\boldsymbol{W}_{l+1}$ in engaged mode. Contrarily, in mirror mode, the feedback neurons mimic the forward neurons, and the feedback weights $\boldsymbol{B}_{l+1}$ are updated by equation (6). However, in our ASAP algorithm, the forward weights and feedback weights can be updated simultaneously since the activity of the mimicking neurons (e.g., neurons in layers $l+1$ and $l+2$ in Fig. 4b) is not used for updating feedback weights. In addition, our ASAP algorithm could be modified to only use neuron-specific signals without introducing mimicry. See Appendix B for more details.

## 5 Experiments

To verify the effectiveness of the proposed ASAP algorithm, we measured its training performance for classification tasks on different datasets. We also implemented the backpropagation and other biologically plausible algorithms with and without bidirectional connections for comparisons. Since we aim at training deep convolutional networks using biologically plausible algorithms, we choose AlexNet [35], ResNet-18, and ResNet-34 [36] for experiments. The target models are trained on MNIST [37], SVHN [38], CIFAR-10, CIFAR-100 [39], and Tiny ImageNet [40] datasets for image classification. In training, we used the stochastic gradient descent (SGD) with the momentum method [41]. In addition, the weight decay method [26] was adopted, and the learning rate was adjusted through cosine annealing [42].

In all experiments, the block size $k$ was set to 2, but we also experimented with $k = 4$ for ResNet-34. Since fully-connected layers can be trained sufficiently well with the Feedback Alignment, ASAP was applied only in convolutional layers, while fully-connected layers are trained using Feedback Alignment. In ASAP, we update the weights of the layers in a block using the shared activations. However, when the channel size increases as it passes through a convolutional layer or the feature map size decreases after a maxpool or convolutional layer, the shared activations cannot be used for weight changes in later layers because of the difference in dimensions. In this case, we apply maxpool to the shared activation when activation map sizes differ and concatenate along the channel dimension when channel sizes vary, to match its dimension with that of the corresponding layer. The input

activation of the first layer, which is the network's input, has a much lower number of channels than the following layers, so the input activation cannot be shared with later layers for training. Therefore, we use random fixed weights in the first layer and apply ASAP to the remaining convolutional layers. More details on the experiments are provided in Appendix C.

Table 1: Test accuracy of BP, KP, FA, DFA, and ASAP (k=2) in classification task

| Dataset | Model | BP | KP | FA | DFA | ASAP |
|---|---|---|---|---|---|---|
| MNIST | AlexNet | 99.59 | 99.55 | 99.14 | 99.28 | 99.32 |
| SVHN | AlexNet | 94.64 | 93.3 | 82.21 | 87.42 | 88.04 |
| | ResNet-18 (nsc*) | 96.14 | 95.84 | 84.91 | 85.64 | 93.17 |
| | ResNet-18 | 96.29 | 96.13 | 85.08 | 85.56 | 94.86 |
| CIFAR-10 | AlexNet | 90.58 | 79.23 | 67.92 | 73.85 | 78.25 |
| | ResNet-18 (nsc*) | 94.70 | 94.65 | 71.38 | 75.46 | 83.44 |
| | ResNet-18 | 94.93 | 94.76 | 72.47 | 76.0 | 92.19 |
| | ResNet-34 | 95.18 | 94.54 | 66.99 | 73.02 | 93.97 |
| CIFAR-100 | AlexNet | 63.61 | 47.43 | 33.32 | 35.38 | 46.99 |
| | ResNet-18 (nsc*) | 77.34 | 74.98 | 37.11 | 37.48 | 51.72 |
| | ResNet-18 | 77.74 | 74.51 | 38.15 | 38.51 | 68.86 |
| | ResNet-34 | 78.41 | 75.0 | 33.01 | 35.6 | 72.81 |
| Tiny ImageNet | ResNet-18 | 60.13 | 58.14 | 20.54 | 24.07 | 48.46 |
| | ResNet-34 | 62.63 | 59.45 | 16.86 | 21.1 | 52.25 |

*no shortcut

**Good performance on deep convolutional networks**    Table 1 summarizes the performance of training algorithms on different datasets and models. In summary, ASAP outperforms other biologically plausible algorithms without bidirectional connections (FA and DFA) in all cases, and shows very competitive performance especially on deep convolutional networks. However, it exhibits some performance degradation compared to KP that employs bidirectional connections. When training the relatively simple AlexNet on the smallest MNIST dataset, FA, DFA, and ASAP all achieve good training performances and closely match BP. For more complex datasets such as CIFAR-10, CIFAR-100, and Tiny ImageNet, ASAP shows significantly better test accuracy than the other algorithms without bidirectional connections. When compared to BP, ASAP shows some performance degradation, but interestingly, the performance gap between ASAP and BP is considerably smaller for deeper convolutional networks such as ResNet-18 and ResNet-34. ASAP only shows a 1.43% accuracy drop for ResNet-18 trained on SVHN, while FA and DFA show more than 10% performance degradation. This trend is confirmed more clearly with complex datasets: for CIFAR-10, ResNet-18 and ResNet-34 trained with ASAP demonstrate over 90% test accuracy with less than 3% performance drop compared to BP, whereas FA and DFA only show up to 76% accuracy. For CIFAR-100, ASAP achieves 68.86% and 72.81% test accuracy for ResNet-18 and ResNet-34, respectively, translating to less than 9% performance degradation over BP. Contrarily, FA and DFA exhibit a much lower test accuracy in the range of 33-39%. For Tiny ImageNet, ASAP achieves a competitive performance of 48.46% and 52.25% for ResNet-18 and ResNet-34, while FA and DFA exhibit only 16-24% test accuracy. Generally, as a convolutional network gets deeper, its performance gradually improves [43]. Nevertheless, the test accuracy of FA and DFA rarely improved by using more convolutional layers in the model. On the other hand, ASAP successfully exploits deep network structures and continues to improve test accuracy when having more layers in the model. Through these experiments, we can confirm that ASAP can train deep convolutional networks on complex datasets such as CIFAR-100 and Tiny ImageNet.

**Effect of shortcut**    ResNet models employ shortcut connections to prevent vanishing and exploding gradients, enabling reliable training on deep convolutional networks [36]. For this reason, when training ResNet-18 through BP, KP, FA, and DFA, having shortcut connections generally increases test accuracy in SVHN, CIFAR-10, and CIFAR-100. Interestingly, the performance gap appears larger for ASAP in all datasets. This is because ASAP uses the activation of a lower layer to update weights. In the absence of shortcuts, the activation of a lower layer is only used to update the weights of ASAP and is not used at all in BP. However, if there is a shortcut, the activation of a lower layer is added to the output activation of the previous layer for weight update in BP. Therefore, the direction

of weight changes of ASAP and BP becomes more similar when there is a shortcut. See Appendix E for more details.

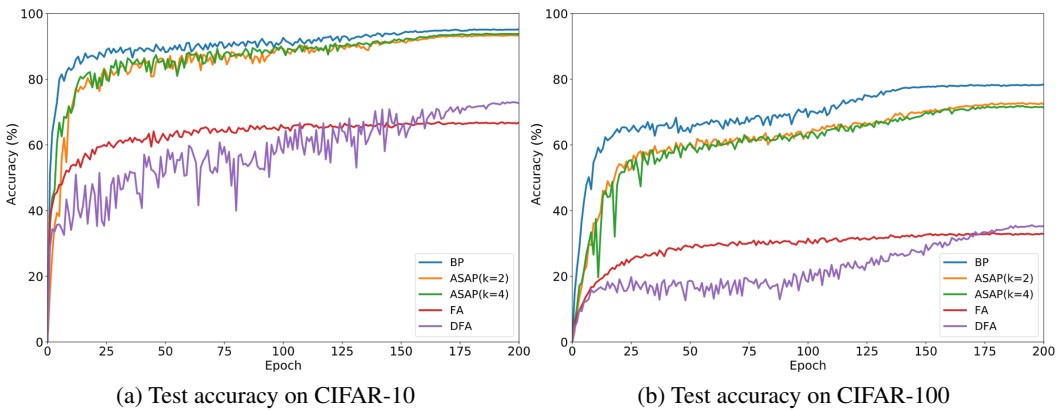

(a) Test accuracy on CIFAR-10          (b) Test accuracy on CIFAR-100

Figure 5: Training performance comparisons for ResNet-34

**Effect of block size**   ASAP removes bidirectional connections by sharing activations across the layers in a block, so the block size itself does not determine biological plausibility. As the block size increases, the number of layers updated using the same shared activations also increases. In other words, the amount of activation required to perform weight updates in training decreases due to data reuse. Therefore, if we perform Activation Sharing with a larger block size $k$, it is possible to reduce memory access when implemented in hardware. We compare the test accuracy of ResNet-34 for $k = 2$ and $k = 4$ in Fig. 5. On CIFAR-10, there was no performance degradation when using a larger block size ($k = 4$). The test accuracy is 93.97% when $k = 2$ and 93.9% when $k = 4$, while BP shows 95.18% test accuracy. On CIFAR-100, the test accuracy is 71.93% when $k = 4$ and 72.81% when $k = 2$, which translates to 0.88% degradation due to increasing block size from 2 to 4. However, using the block size of 4 considerably reduces memory access because only about 25% of activations are used compared to BP and 50% compared to $k = 2$ for weight updates (see Appendix F for details).

## 6 Discussion

It has been shown that our algorithm solves the weight transport problem without bidirectional connections by updating forward and feedback weights using shared activation. While ASAP still exhibits some performance degradation compared to BP and KP, it achieves competitive performance in deep convolutional networks on complex datasets without using bidirectional connections. Of course, our method is not completely biologically plausible. Unlike the brain, ASAP does not use spikes to propagate signals. Also, our algorithm still requires a paired error pathway (i.e., each forward neuron is paired with a corresponding feedback neuron), which has no biological evidence yet.

Nevertheless, our study is meaningful in that it shows that it is possible to train deep neural networks with approximate activation, while previous biological learning methods require accurate neuronal activity. This approximate learning rule not only eliminates bidirectional connections from algorithms that assume a paired error pathway, but could also be applied to algorithms assuming a single pathway to alleviate the constraint of accurate neural activity, which will be investigated further in future work.

When implementing deep learning in hardware, a large number of MAC operations and external memory access are the processing bottlenecks [44]. In backpropagation, activations calculated in the forward path are required in the feedback path to update weights. For large models, it is impossible to store all the activations on a chip; hence, it is inevitable to store the activations externally and load them back to the chip later. Contrarily, in our algorithm, not all the activations are required for weight updates due to the approximate forward path. Therefore, the ASAP algorithm is a good candidate for efficient hardware implementation since it could greatly reduce memory access overheads.

## Acknowledgments

This work was supported by the National Research Foundation of Korea (Grant No. NRF-2019R1C1C1004927) and the KIST Institutional Program (Project No. 2E30610-20-058).

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
