## Appendices

## A Forward Alignment

We consider a linear network $\boldsymbol{h} = \boldsymbol{Ax}$, $\boldsymbol{y} = \boldsymbol{Wh}$ which generates an output $\boldsymbol{y}$ from an input $\boldsymbol{x}$, where the input $\boldsymbol{x}$, the hidden layer output $\boldsymbol{h}$, and the output $\boldsymbol{y}$ are all column vectors and the number of neurons (i.e., dimension of each vector) follows $n_h > n_x > n_y$. $\boldsymbol{A}$ and $\boldsymbol{W}$ denote the weights of the first layer and the second layer, respectively. The purpose of the network is to make a desired output $\mathring{\boldsymbol{y}} = \boldsymbol{Tx}$ where $\boldsymbol{T}$ is a linear target function. In this section, we show that the Forward Alignment algorithm, which uses approximate activations $\tilde{\boldsymbol{h}} = \boldsymbol{Cx}$, where $\boldsymbol{C}$ denotes random fixed weights, instead of $\boldsymbol{h}$ to calculate the change of the second layer weight ($\Delta \boldsymbol{W}$), can also reduce loss to zero in specific conditions. We will also show experimentally that the forward weights and the random fixed weights are aligned in simple linear and nonlinear networks when Forward Alignment is applied.

### A.1 Mathematical proof that loss converges to zero in Forward Alignment

For proof, we define

$$\boldsymbol{E} = \boldsymbol{T} - \boldsymbol{WA} \tag{25}$$

to simply express the error as $\boldsymbol{e} = \boldsymbol{Ex}$. We can express the weight changes of layers as

$$\Delta \boldsymbol{W} = \eta \, \boldsymbol{e} \, \tilde{\boldsymbol{h}}^T = \eta \, \boldsymbol{E} \, \boldsymbol{x} \, \boldsymbol{x}^T \, \boldsymbol{C}^T \tag{26}$$

$$\Delta \boldsymbol{A} = \eta \, \boldsymbol{W}^T \boldsymbol{e} \, \boldsymbol{x}^T = \eta \, \boldsymbol{W}^T \boldsymbol{E} \, \boldsymbol{x} \, \boldsymbol{x}^T \tag{27}$$

where $\eta$ denotes the learning rate.

Assuming that we update the weight matrices A and W only once after providing many training samples by the amount of weight change averaged over all training samples, the weight changes are expressed by

$$\Delta \boldsymbol{W} = \eta \, [\boldsymbol{E} \, \boldsymbol{x} \, \boldsymbol{x}^T \, \boldsymbol{C}^T] = \eta \, \boldsymbol{E} \, \boldsymbol{C}^T \tag{28}$$

$$\Delta \boldsymbol{A} = \eta \, [\boldsymbol{W}^T \boldsymbol{E} \, \boldsymbol{x} \, \boldsymbol{x}^T] = \eta \, \boldsymbol{W}^T \boldsymbol{E} \tag{29}$$

where [] denotes an expected value. We assume that the elements of input $\boldsymbol{x}$ are i.i.d. random variables with a mean of 0 and a standard variation of 1. Therefore, $[\boldsymbol{xx}^T] = \boldsymbol{I}$ where $\boldsymbol{I}$ is the identity matrix. If we use a very low learning rate, discrete update process can be approximated to continuous time dynamics as below.

$$\dot{\boldsymbol{W}} = \boldsymbol{E} \, \boldsymbol{C}^T \tag{30}$$

$$\dot{\boldsymbol{A}} = \boldsymbol{W}^T \boldsymbol{E} \tag{31}$$

where $\dot{a}$ denotes $da/dt$. By combining equations (30) and (31), we have

$$\dot{\boldsymbol{A}} \boldsymbol{C}^T = \boldsymbol{W}^T \boldsymbol{E} \boldsymbol{C}^T = \boldsymbol{W}^T \dot{\boldsymbol{W}} \tag{32}$$

$$\boldsymbol{A} \boldsymbol{C}^T = \int \boldsymbol{W}^T \dot{\boldsymbol{W}} + \boldsymbol{k} = \frac{1}{2} \boldsymbol{W}^T \boldsymbol{W} + \boldsymbol{k} \tag{33}$$

where $\boldsymbol{k}$ is a constant. We then propose a theorem below.

**Theorem.** *If the weight changes are given by*

$$\dot{W} = E\,C^T$$

$$\dot{A} = W^T E$$

*where $k$ in equation (33) is zero and $C^+ C = I$, then*

$$\lim_{t \to 0} E = 0$$

Here, $C^+$ denotes the Moore-Penrose pseudoinverse of $C$. The condition $C^+ C = I$ can be satisfied when the columns of $C$ are independent and $C$ has at least as many rows as columns (i.e., $n_h > n_x$). The condition $k = 0$ can be satisfied when $AC^T = \frac{1}{2} W^T W$.

Inspired by Lillicrap et al. [25] who employed Lyapunov's method, we can prove the theorem using Barbalat's lemma.

**Lemma 1** (Barbalat's Lemma). *When $V$ satisfies the following conditions:*

*1. $V$ is lower bounded,*

*2. $\dot{V}$ is negative semi-definite, and*

*3. $\dot{V}$ is uniformly continuous in time if $\ddot{V}$ is finite,*

*then $\dot{V} \to 0$ as $t \to \infty$.*

Here, we define the matrix $V$ as

$$V = tr(EC^T C E) \tag{34}$$

To prove the theorem, we first prove that $\dot{V}$ converges to zero using Barbalat's lemma. We then prove the loss $\|E\|$ converges to zero.

**Lemma 2.** *$V$ is lower bounded.*

*Proof.* $V = \|EC^T\|$ because $E$ and $C$ are real valued. Therefore, $V$ is lower bounded (i.e., $V \geq 0$). $\qquad\square$

**Lemma 3.** *$\dot{V}$ is negative semi-definite.*

*Proof.*

$$\frac{d}{dt} tr(EC^T C E^T) = tr(\dot{E}C^T C E^T) + tr(EC^T C \dot{E}^T)$$

$$= 2tr(\dot{E}C^T C E^T)$$

$$= 2tr((-\dot{W}A - W\dot{A})C^T C E^T)$$

$$= -2tr(\dot{W}AC^T C E^T) - 2tr(W\dot{A}C^T C E^T)$$

$$= -2tr(EC^T AC^T C E^T) - 2tr(WW^T EC^T C E^T)$$

by using equations (30) and (31). Then,

$$2tr(EC^T AC^T C E^T) = tr(EC^T AC^T C E^T) + tr(EC^T C A^T C E^T)$$

$$= \frac{1}{2}(tr(EC^T W^T W C E^T) + tr(EC^T W^T W C E^T))$$

$$= tr(EC^T W^T W C E^T)$$

Consequently, we attain

$$\frac{d}{dt}tr(\boldsymbol{E}\boldsymbol{C}^T\boldsymbol{C}\boldsymbol{E}^T) = -2tr(\boldsymbol{E}\boldsymbol{C}^T\boldsymbol{A}\boldsymbol{C}^T\boldsymbol{C}\boldsymbol{E}^T) - 2tr(\boldsymbol{W}\boldsymbol{W}^T\boldsymbol{E}\boldsymbol{C}^T\boldsymbol{C}\boldsymbol{E}^T)$$
$$= -tr(\boldsymbol{E}\boldsymbol{C}^T\boldsymbol{W}^T\boldsymbol{W}\boldsymbol{C}\boldsymbol{E}^T) - 2tr(\boldsymbol{W}^T\boldsymbol{E}\boldsymbol{C}^T\boldsymbol{C}\boldsymbol{E}^T\boldsymbol{W})$$

because the commutative law holds for trace operation. Since each term is the inner product of a vector with itself, $\dot{V}$ is negative semi-definite.

$\square$

**Lemma 4.** $\boldsymbol{W}$ *is bounded.*

*Proof.* we define $s$ as

$$s = tr(\boldsymbol{W}\,\boldsymbol{W}^T)$$

Then,

$$\dot{s} = 2tr(\boldsymbol{E}\boldsymbol{C}^T\boldsymbol{W}^T)$$
$$= 2tr((\boldsymbol{T} - \boldsymbol{W}\boldsymbol{A})\boldsymbol{C}^T\boldsymbol{W}^T)$$
$$= 2tr(\boldsymbol{T}\boldsymbol{C}^T\boldsymbol{W}^T) - 2tr(\frac{1}{2}\boldsymbol{W}\boldsymbol{W}^T\boldsymbol{W}\boldsymbol{W}^T)$$
$$= 2tr(\boldsymbol{T}\boldsymbol{C}^T\boldsymbol{W}^T) - tr(\boldsymbol{W}\boldsymbol{W}^T\boldsymbol{W}\boldsymbol{W}^T)$$

Here, $\boldsymbol{W}\boldsymbol{W}^T$ is diagonalizable because it is an $n_o \times n_o$ symmetric matrix. Then,

$$s < n_o\lambda$$

where $\lambda$ denotes the dominant eigenvalue of $\boldsymbol{W}\boldsymbol{W}^T$. Therefore,

$$tr(\boldsymbol{W}\boldsymbol{W}^T\boldsymbol{W}\boldsymbol{W}^T) = \|\boldsymbol{W}\boldsymbol{W}^T\|$$
$$\geq \lambda^2$$
$$\geq (\frac{s}{n_o})^2$$

Consequently, we can attain

$$\dot{s} = 2tr(\boldsymbol{T}\boldsymbol{C}^T\boldsymbol{W}^T) - tr(\boldsymbol{W}\boldsymbol{W}^T\boldsymbol{W}\boldsymbol{W}^T)$$
$$\leq 2tr(\boldsymbol{T}\boldsymbol{C}^T\boldsymbol{W}^T) - (\frac{s}{n_o})^2$$

Using Caushy-Schwarz inequality, it can be shown that

$$tr(\boldsymbol{T}\boldsymbol{C}^T\boldsymbol{W}^T)^2 \leq tr(\boldsymbol{T}\boldsymbol{C}^T\boldsymbol{C}\boldsymbol{T}^T)\,tr(\boldsymbol{W}^T\boldsymbol{W}) = s\|\boldsymbol{T}\boldsymbol{C}^T\|^2$$

Therefore, when $s > \|\boldsymbol{T}\boldsymbol{C}^T\|^2$, we can get the inequality

$$\dot{s} < 2s - \frac{s^2}{n_o^2}$$

Hence,

$$\dot{s} < 0$$

when $s > \boldsymbol{T}\boldsymbol{C}^T$ and $s > 2n_o$. Using this relationship, we confirm that

$$s \leq \boldsymbol{T}\boldsymbol{C}^T + 2n_o$$

at any time $t$. Therefore, $\boldsymbol{W}$ is bounded.

$\square$

**Lemma 5.** $\ddot{V}$ *is finite.*

*Proof.* Using $\dot{V}$ obtained in Lemma 3, $\ddot{V}$ is calculated as

$$\begin{aligned}
\ddot{V} &= \frac{d}{dt}[tr(\boldsymbol{E}\boldsymbol{C}^T\boldsymbol{W}^T\boldsymbol{W}\boldsymbol{C}\boldsymbol{E}^T) - 2tr(\boldsymbol{W}^T\boldsymbol{E}\boldsymbol{C}^T\boldsymbol{C}\boldsymbol{E}^T\boldsymbol{W})] \\
&= -2tr(\dot{\boldsymbol{E}}\boldsymbol{C}^T\boldsymbol{W}^T\boldsymbol{W}\boldsymbol{C}\boldsymbol{E}^T) - 2tr(\boldsymbol{E}\boldsymbol{C}^T\dot{\boldsymbol{W}}^T\boldsymbol{W}\boldsymbol{C}\boldsymbol{E}^T) \\
&\quad - 4tr(\dot{\boldsymbol{W}}^T\boldsymbol{E}\boldsymbol{C}^T\boldsymbol{C}\boldsymbol{E}^T\boldsymbol{W}) - 4tr(\boldsymbol{W}^T\dot{\boldsymbol{E}}\boldsymbol{C}^T\boldsymbol{C}\boldsymbol{E}^T\boldsymbol{W}) \\
&= -2tr((-\boldsymbol{W}\dot{\boldsymbol{A}} - \dot{\boldsymbol{W}}\boldsymbol{A})\boldsymbol{C}^T\boldsymbol{W}^T\boldsymbol{W}\boldsymbol{C}\boldsymbol{E}^T) - 2tr(\boldsymbol{E}\boldsymbol{C}^T\dot{\boldsymbol{W}}^T\boldsymbol{W}\boldsymbol{C}\boldsymbol{E}^T) \\
&\quad - 4tr(\dot{\boldsymbol{W}}^T\boldsymbol{E}\boldsymbol{C}^T\boldsymbol{C}\boldsymbol{E}^T\boldsymbol{W}) - 4tr(\boldsymbol{W}^T(-\boldsymbol{W}\dot{\boldsymbol{A}} - \dot{\boldsymbol{W}}\boldsymbol{A})\boldsymbol{C}^T\boldsymbol{C}\boldsymbol{E}^T\boldsymbol{W}) \\
&= 2tr(\boldsymbol{W}\boldsymbol{W}^T\boldsymbol{E}\boldsymbol{C}^T\boldsymbol{W}^T\boldsymbol{W}\boldsymbol{C}\boldsymbol{E}^T) + 2tr(\boldsymbol{E}\boldsymbol{C}^T\boldsymbol{A}\boldsymbol{C}^T\boldsymbol{W}^T\boldsymbol{W}\boldsymbol{C}\boldsymbol{E}^T) - 2tr(\boldsymbol{E}\boldsymbol{C}^T\boldsymbol{E}\boldsymbol{C}^T\boldsymbol{W}\boldsymbol{C}\boldsymbol{E}^T) \\
&\quad - 4tr(\boldsymbol{E}\boldsymbol{C}^T\boldsymbol{E}\boldsymbol{C}^T\boldsymbol{C}\boldsymbol{E}^T\boldsymbol{W}) + 4tr(\boldsymbol{W}^T\boldsymbol{W}\boldsymbol{W}^T\boldsymbol{E}\boldsymbol{C}^T\boldsymbol{C}\boldsymbol{E}^T\boldsymbol{W}) + 4tr(\boldsymbol{W}^T\boldsymbol{E}\boldsymbol{C}^T\boldsymbol{A}\boldsymbol{C}^T\boldsymbol{C}\boldsymbol{E}^T\boldsymbol{W})
\end{aligned}$$

Here, $\ddot{V}$ can be expressed using $\boldsymbol{W}$, $\boldsymbol{C}$, $\boldsymbol{E}$, and $\boldsymbol{A}\boldsymbol{C}^T$. By Lemma 4, $\boldsymbol{W}$ is bounded and $\boldsymbol{C}$ is a random fixed matrix. Lemma 2 shows that $V$ is upper bounded by zero while $\dot{V} \leq 0$ by Lemma 3. Therefore, $V$ must converge to some value and then $\boldsymbol{E}$ must be bounded because $V = tr(\boldsymbol{E}\boldsymbol{C}\boldsymbol{C}^T\boldsymbol{E})$ in equation (34). Also, $\boldsymbol{A}\boldsymbol{C}^T$ is bounded because $\boldsymbol{A}\boldsymbol{C}^T = \frac{1}{2}\boldsymbol{W}^T\boldsymbol{W}$ by equation (33). Consequently, all terms of $\ddot{V}$ are bounded, which means $\ddot{V}$ is finite. $\square$

According to the Lemmas above, the conditions of Barbalat's lemma are satisfied and $\dot{V} \to 0$ as $t \to \infty$. By the equations in the proof of Lemma 3,

$$\dot{V} = -\|\boldsymbol{E}\boldsymbol{C}^T\boldsymbol{W}^T\|^2 - 2\|\boldsymbol{W}^T\boldsymbol{E}\boldsymbol{C}^T\|^2 \tag{35}$$

where both addends of $\dot{V}$ have the same sign. Therefore, we can conclude that $\boldsymbol{W}^T\boldsymbol{E}\boldsymbol{C}^T = 0$. Since $\boldsymbol{C}$ is constant and has left pseudoinverse, $\boldsymbol{A} = \boldsymbol{W}^T\boldsymbol{E} = 0$. By equation (33), $\boldsymbol{A}\boldsymbol{C}^T = \frac{1}{2}\boldsymbol{W}^T\boldsymbol{W}$ and $\boldsymbol{W}^T\boldsymbol{W}$ is constant because $\boldsymbol{A}$ and $\boldsymbol{C}$ are constant. By using $\boldsymbol{W}^T\boldsymbol{E}\boldsymbol{C}^T = 0$, we can have

$$\boldsymbol{W}^T\boldsymbol{E}\boldsymbol{C}^T = \boldsymbol{W}^T(\boldsymbol{T} - \boldsymbol{W}\boldsymbol{A})\boldsymbol{C}^T = 0 \tag{36}$$

Here, $\boldsymbol{W}^T\boldsymbol{T}\boldsymbol{C}^T = \boldsymbol{W}^T\boldsymbol{W}\boldsymbol{A}\boldsymbol{C}^T$ and then $\boldsymbol{W}^T\boldsymbol{T}\boldsymbol{C}^T$ is constant because $\boldsymbol{W}^T\boldsymbol{W}$, $\boldsymbol{A}$, and $\boldsymbol{C}^T$ are all constant. By differentiating $\boldsymbol{W}^T\boldsymbol{T}\boldsymbol{C}^T$, we can get

$$\dot{\boldsymbol{W}}^T\boldsymbol{T}\boldsymbol{C}^T = \boldsymbol{C}\boldsymbol{E}^T\boldsymbol{T}\boldsymbol{C}^T = 0 \tag{37}$$

because $\dot{\boldsymbol{W}} = \boldsymbol{E}\boldsymbol{C}^T$ by equation (30). Since $\boldsymbol{C}$ has left pseudoinverse, we can modify the equation (37) to

$$0 = \boldsymbol{C}\boldsymbol{E}^T\boldsymbol{T}\boldsymbol{C}^T = \boldsymbol{C}^+\boldsymbol{C}\boldsymbol{E}^T\boldsymbol{T}\boldsymbol{C}^T(\boldsymbol{C}^+)^T = \boldsymbol{E}^T\boldsymbol{T} = \boldsymbol{T}^T\boldsymbol{E} \tag{38}$$

By equation (25), $\boldsymbol{E}^T\boldsymbol{E} = (\boldsymbol{T}^T - \boldsymbol{A}^T\boldsymbol{W}^T)\boldsymbol{E}$. Since $\boldsymbol{W}^T\boldsymbol{E}$ and $\boldsymbol{T}^T\boldsymbol{E}$ are zero, $\boldsymbol{E}^T\boldsymbol{E}$ becomes zero. In conclusion, $tr(\boldsymbol{E}^T\boldsymbol{E}) = \|\boldsymbol{E}\| = 0$ and hence the loss of the network $\boldsymbol{E}$ converges to zero.

## A.2    Weight alignment in Forward Alignment

In the section above, we demonstrated that the Forward Alignment algorithm has the potential to train neural networks under certain conditions. In this section, we experimentally demonstrate the algorithm's capability of neural network training. We first observe whether an accuracy similar to BP could be obtained, and then we verify that the forward weights and the random fixed weights are aligned. Here we use two target networks having a 10-10-10-10 structure: a linear network without batch normalization [45] and ReLU function, and a nonlinear network including all units. The two networks are trained on the MNIST dataset for 200 epochs. The batch size is set to 50, and the learning late is 1e-4. Other learning methods and hyperparameters are the same as those presented in Appendix C.

In this experiment, we obtained a test accuracy of 90.93% in the linear network and 93.03% in the nonlinear network, which closely matches the backpropagation with 92.39% and 95.01% accuracy on the same networks. Furthermore, Fig. 6 confirms that the forward weights and the random fixed weights are partially aligned, which is similar to the partial alignment between the feedback weights and the random fixed weights observed in the Feedback Alignment algorithm [8]. In the figure, layer 1 denotes the first $10 \times 10$ layer and layer 2 denotes the second $10 \times 10$ layer. In addition, Fig. 6b confirms that the Forward Alignment algorithm can train not only linear networks but also nonlinear networks. Interestingly, even though the weights are not well aligned in the nonlinear network compared to the linear network, the test accuracy is higher for the nonlinear network. It was experimentally shown that the alignment of weights in the Feedback Alignment was not directly related to training performance by Moskovitz et al. [11]. Therefore, we suspect that the alignment of weights is not the cause of learning, but rather a phenomenon observed when a network is trained using Forward Alignment or Feedback Alignment.

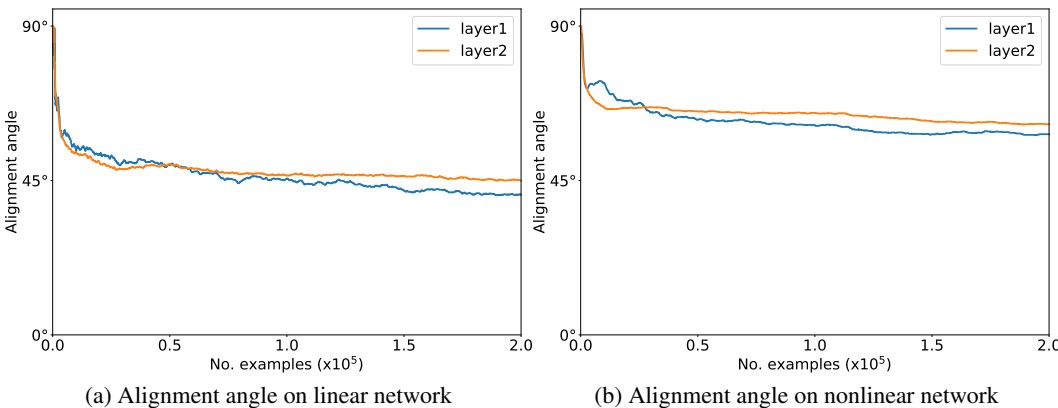

(a) Alignment angle on linear network          (b) Alignment angle on nonlinear network

Figure 6: Weight alignment in Forward Alignment.

## B    Activation Sharing in Feedback Path: the algorithm using neuron-specific signals without mimicry

We show that our ASAP algorithm can be implemented biologically by mimicking shared activation in Section 4.2. However, if we modify the algorithm in a way that the shared activation is used only for updating feedback weights, we could avoid using synapse-specific signals even without introducing mimicry in the algorithm. In other words, forward weights are updated using pre-synaptic activations as in conventional algorithms, while feedback weights are updated using shared activations. The learning process for $k = 2$ is expressed by the equations below.

$$\Delta \boldsymbol{W}_{l+2} = -\eta \, \boldsymbol{\delta}_{l+2} \, \boldsymbol{h}_{l+1}^T \tag{39}$$

$$\Delta \boldsymbol{W}_{l+3} = -\eta \, \boldsymbol{\delta}_{l+3} \, \boldsymbol{h}_{l+2}^T \tag{40}$$

$$\Delta \boldsymbol{B}_{l+2} = -\eta \, \boldsymbol{\delta}_{l+2} \, \boldsymbol{h}_l^T \tag{41}$$

$$\Delta \boldsymbol{B}_{l+3} = -\eta \, \boldsymbol{\delta}_{l+3} \, \boldsymbol{h}_l^T \tag{42}$$

We name this algorithm Activation Sharing in Feedback Path (ASFP). Unlike ASAP, ASFP uses pre-synaptic neuron activity ($\boldsymbol{h}_{l+1}$ and $\boldsymbol{h}_{l+2}$) for forward weight updates ($\Delta \boldsymbol{W}_{l+2}$ and $\Delta \boldsymbol{W}_{l+3}$). As a result, we no longer need mimicry in the forward path as shown in Fig. 7b.

Table 2: Test accuracy of BP, ASAP, ASFP and ASFP with local classifier on CIFAR-10

| Architecture | BP | ASAP | ASFP | ASFP + LC* |
|---|---|---|---|---|
| ResNet-18 | 94.93 | 92.19 | 61.45 | 88.46 |
| ResNet-34 | 95.18 | 93.97 | 31.7 | 89.28 |

*Local Classifier

We experimented with the ASFP algorithm with $k = 2$ on CIFAR-10 dataset under the same test environments described in Appendix C. Experimental results are displayed in Table 2. Unfortunately, the ASFP algorithm exhibits significant performance degradation when applied to deep neural networks. We speculate that this performance degradation is due to the difference in the weight update directions of the forward and feedback weights (i.e., $\Delta \boldsymbol{W} \neq \Delta \boldsymbol{B}$) as shown in equations (39)-(42)). To mitigate performance drop, we propose to adopt local learning [46] in the algorithm. Specifically, a local classifier consisting of average pool layers whose output dimension is 2×2 and fully-connected layers is added at the end of each block. Like ASAP, fully-connected layers are trained using Feedback Alignment. This ASFP with local classifiers (ASFP+LC) algorithm achieves a competitive performance of 88-89%. Although it exhibits lower test accuracy compared to BP and ASAP, it outperforms FA, DFA, and other local learning algorithms [46, 47, 48] (see Appendix D.4 for comparisons with local learning algorithms). We speculate that local learning minimizes the difference in the weight update directions in the ASFP algorithm. In summary, applying activation sharing only to the feedback path effectively removes the need for synapse-specific signals without mimicry with the aid of local learning.

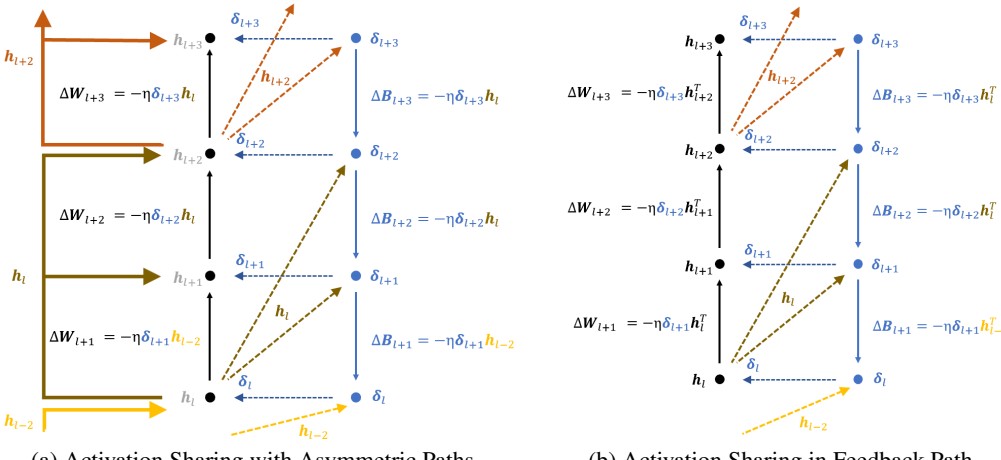

(a) Activation Sharing with Asymmetric Paths      (b) Activation Sharing in Feedback Path

Figure 7: Overview of ASAP and ASFP.

## C    Experimental details

**Datasets**    The MNIST dataset [37] consists of 70,000 $28 \times 28$ grayscale images. 60,000 images are used as the training set, and 10,000 images are used as the test set. The SVHN dataset [38] has $32 \times 32$ RGB images of digits, which are divided into 73,257 training images and 26,032 test images. The CIFAR-10 dataset [39] includes 60,000 $32 \times 32$ RGB images of 10 classes. It is split into 50,000

training images and 10,000 test images. The CIFAR-100 dataset [39] also consists of 60,000 $32 \times 32$ RGB images of 100 classes, which are divided into 50,000 training samples and 10,000 test samples. The Tiny ImageNet dataset [40] is a small subset of the ImageNet dataset [40]. It consists of 100,000 $64 \times 64$ RGB training images and 10,000 $64 \times 64$ test images of 200 classes. We normalized the images using the channel mean and standard variation. For data augmentation, random crop and random horizontal flip methods are applied in CIFAR-10, CIFAR-100 and Tiny Imagenet datasets.

**Architecture and hyperparameters**   For experiments, we implemented AlexNet [35], ResNet-18, ResNet-18 without shortcuts, and ResNet-34 [36] using PyTorch library on Nvidia GeForce GTX 1080 Ti GPUs. The networks are trained using the stochastic gradient descent with the momentum optimizer [41] for 200 epochs. The AlexNet has three convolutional layers with channel sizes of 64, 128, and 256, respectively, $3 \times 3$ kernel size, 1 stride, and 2 padding. After the convolutional layer, a batch normalization layer [45] and ReLU activation follow. Then, a maxpool layer with the kernel size of 2 and the stride of 2 is employed. The classifier is implemented using three fully-connected layers with the structure of 1024-1024-10. The ResNet-18 consists of convolutional layers with 64, 128, 256, and 512 channel sizes, $3 \times 3$ kernel, 1 stride, and 2 padding. However, when the channel size changes, convolutional layers with 2 stride are used. The batch normalization and ReLU function are also employed in ResNet-18. Finally, an adaptive average pooling layer and a linear layer are adopted in the classifier. More details such as the number of convolutional layers and the arrangement of shortcuts could be found in [36]. The Kaiming initialization [49] is applied to all convolutional layers. We set the momentum to 0.9 and the L2 weight decay ratio to 5e-4. The batch size is 128, and the cosine learning rate annealing [42] is applied. Since the optimal learning rate differs in various learning methods, we use different initial learning rates displayed in Table 3.

Table 3: Learning rates used in experiments.

| Dataset | Model | BP | KP | FA | DFA | ASAP |
|---|---|---|---|---|---|---|
| MNIST | AlexNet | 3e-2 | 3e-2 | 1e-4 | 1e-4 | 1e-4 |
| SVHN | AlexNet | 3e-2 | 3e-2 | 1e-4 | 1e-4 | 1e-4 |
| | ResNet-18 (nsc*) | 3e-2 | 3e-2 | 3e-3 | 3e-3 | 3e-3 |
| | ResNet-18 | 3e-2 | 3e-2 | 3e-3 | 3e-3 | 3e-3 |
| CIFAR-10 | AlexNet | 3e-2 | 3e-2 | 1e-4 | 1e-4 | 1e-4 |
| | ResNet-18 (nsc*) | 3e-2 | 3e-2 | 3e-3 | 3e-3 | 3e-3 |
| | ResNet-18 | 3e-2 | 3e-2 | 3e-3 | 3e-3 | 3e-2 |
| | ResNet-34 | 3e-2 | 3e-2 | 3e-3 | 3e-3 | 3e-2 |
| CIFAR-100 | AlexNet | 3e-2 | 3e-2 | 1e-4 | 1e-4 | 1e-4 |
| | ResNet-18 (nsc*) | 3e-2 | 3e-2 | 3e-3 | 3e-3 | 3e-3 |
| | ResNet-18 | 3e-2 | 3e-2 | 3e-3 | 3e-3 | 3e-2 |
| | ResNet-34 | 3e-2 | 3e-2 | 3e-3 | 3e-3 | 3e-2 |
| Tiny ImageNet | ResNet-18 | 3e-2 | 3e-2 | 3e-3 | 3e-3 | 3e-2 |
| | ResNet-34 | 3e-2 | 3e-2 | 3e-3 | 3e-3 | 3e-2 |

*no shortcut

# D   Additional details of ASAP algorithm

## D.1   Network Architecture

Fig. 8 depicts the ASAP algorithm applied to deep convolutional network. Since the input of the first layer is difficult to use as shared activation due to its small channel size, the first layer is not trained and is left as random fixed weights. Specifically, the input channel size of the first layer is 3, whereas the input channel size of other layers is a multiple of 32 in typical convolutional layers. Therefore, the second layer uses the actual activations to update the weights, not the input of the first layer. For other layers, shared activations are used to update the weights of all layers in each block. The output activations of the the block become the shared activations of the next block. Consequently, all the layers in a block uses the previous block's output activation to update their weights instead of previous layer's activations as in backpropagation.

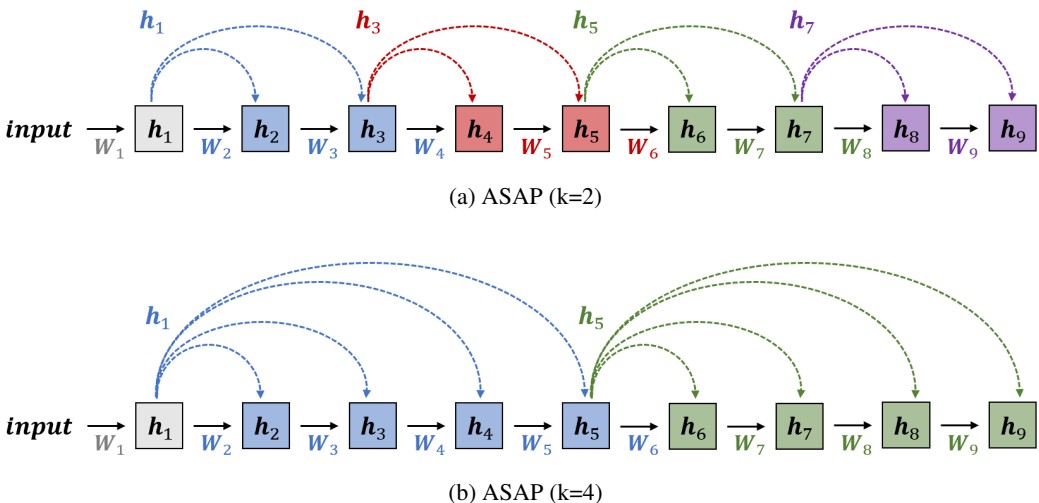

(a) ASAP (k=2)

(b) ASAP (k=4)

Figure 8: Overview of ASAP algorithm.

## D.2 Matching dimension of shared activations

The ASAP algorithm assumes that the shared activations and actual activations have the same dimension. However, there are cases where they have different dimensions. For example, the channel size or feature map size may be changed by a convolutional layer, and the feature map size may be decreased due to pooling. In this case, we have to match the dimensions of the shared activations and actual activations. To change the dimension of the shared activations while retaining information, we experimented with three methods in AlexNet. First, after concatenating multiple clones of the shared activation to match the channel size, maxpool is applied to match the feature map size. Second, we increase the dimension of the shared activations in the same way but apply average pooling. Finally, as with Forward Alignment, an $1 \times 1$ convolutional layer with random fixed weights is applied. Fig. 9 displays that the best performance was obtained when the first method was applied. Nevertheless, the other methods also resulted in relatively good accuracy. While we used the first method (concatenation and max pooling) for experiments in this work, it may be easier to implement ASAP by using a $1 \times 1$ convolutional layer in a network where dimension changes irregularly.

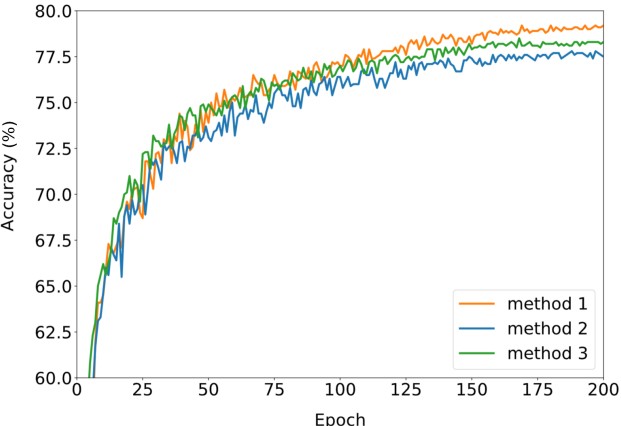

Figure 9: Test accuracy comparisons of dimension matching methods.

Table 4: Test accuracy of diverse weight decay factor ($\lambda$) on CIFAR-10

| Learning rules | $\lambda = 0$ | $\lambda = 5e\text{-}6$ | $\lambda = 5e\text{-}5$ | $\lambda = 5e\text{-}4$ | $\lambda = 5e\text{-}3$ |
|---|---|---|---|---|---|
| BP | 93.56 | 93.76 | 93.82 | 95.18 | 95.12 |
| ASAP | 89.77 | 89.85 | 91.63 | 93.97 | 91.54 |

In Section 3.3, we assume weight decay to prove that the forward and feedback weights converge to identical values. We experimented with BP and ASAP on CIFAR-10 while varying the weight decay factor. Experimental results are provided in Table 4. In both BP and ASAP, weight decay affects training performance, but the performance variation is more significant in ASAP. This is consistent with what we expected, because the weight decay term is necessary in ASAP to reduce the difference in the forward and feedback weights.

## D.4    Comparisons with local learning algorithms

Table 5: Test accuracy of ASAP and local learning algorithms on CIFAR-10

| Architecture | BP | [46] | [47] | [48] | ASAP |
|---|---|---|---|---|---|
| ResNet-18 | 94.93 | 71.41 | 88.12 | 80.76 | 92.19 |
| ResNet-34 | 95.18 | 69.8 | 87.57 | 77.2 | 93.97 |

We compare ASAP to other learning rules employing local errors. Those algorithms are applied to training ResNet-18 and ResNet-34 on the CIFAR-10 dataset, where we used the test environment described in Appendix C. However, for optimal results, the learning rate is set to 5e-3 and the Adam optimizer [50] is used in local learning. Experimental results are displayed in Table 5. Although local learning algorithms achieve good training performance on VGG-like models, they exhibit noticeable performance degradation when applied to ResNet, which was also reported in [47]. We speculate that the skip connections in ResNet models do not improve training performance for local learning as the errors do not backpropagate through layers. On the other hand, our ASAP algorithm takes advantage of shortcut connections, as discussed in Section 5 and Appendix E. Nevertheless, local learning algorithms are meaningful in that it realizes layerwise learning which could be more biologically plausible than end-to-end learning.

## E    Shortcut effect

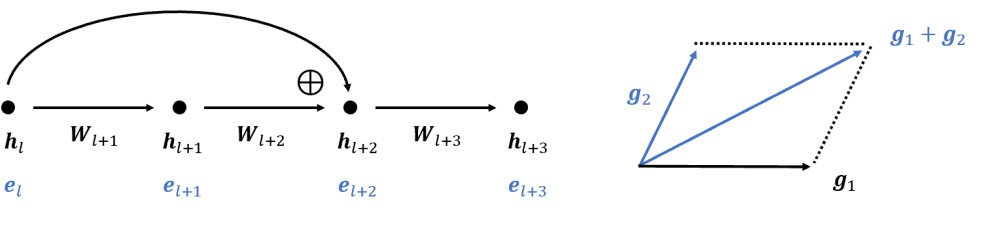

(a) Shortcut in ResNet                    (b) Better alignment due to shortcut effect

Figure 10: Overview of shortcut effect.

In Section 5, we show that the performance of the proposed ASAP algorithm is closer to that of backpropagation when shortcuts are present in ResNet. In this section, we analyze the reason behind this phenomenon. Fig. 10a depicts the residual block used in ResNet. To understand the effect of residual blocks in the ASAP algorithm, we compare the weight changes of layer $l + 3$

in the backpropagation and the ASAP algorithm. If there is no shortcut, the weight change in backpropagation is obtain by

$$h_{l+2} = \phi(W_{l+2}h_{l+1} + b_{l+2}) \tag{43}$$

$$\Delta W_{l+3} = \delta_{l+3}h_{l+2} = \delta_{l+3}\,\phi(W_{l+2}h_{l+1} + b_{l+2}) \tag{44}$$

Similarly, if there is a shortcut, the weight update process is expressed as

$$h_{l+2} = \phi(W_{l+2}h_{l+1} + b_{l+2}) + h_l \tag{45}$$

$$\Delta W_{l+3} = \delta_{l+3}h_{l+2} = \delta_{l+3}\,\phi(W_{l+2}h_{l+1} + b_{l+2}) + \delta_{l+3}h_l \tag{46}$$

On the other hand, the ASAP algorithm updates weights using shared activation $h_l$. Then, the weight changes with and without shortcuts are

$$\Delta W_{l+3} = \delta_{l+3}h_l \tag{47}$$

While the weight changes in the networks with and without residual blocks are all represented by equation (47), the actual value would be different because the value of $\delta_{l+3}$ differs for the two cases due to different network structures. To simplify the equations above, we define $g_1 = \delta_{l+3}\,\phi(W_{l+2}h_{l+1} + b_{l+2})$ and $g_2 = \delta_{l+3}h_l$. Then, equations (44), (46), and (47) are expressed as $g_1$, $g_1 + g_2$, and $g_2$, respectively. This suggests that the weight change of ASAP is closer to that of backpropagation when shortcut connections are present in the network as shown in Fig 10b. In other words, the additional weight change $g_2$ in backpropagation due to shortcuts matches the weight change of ASAP. Consequently, the performance gap between backpropagation and ASAP is reduced when employing shortcuts in the network.

## F  Hardware implementation

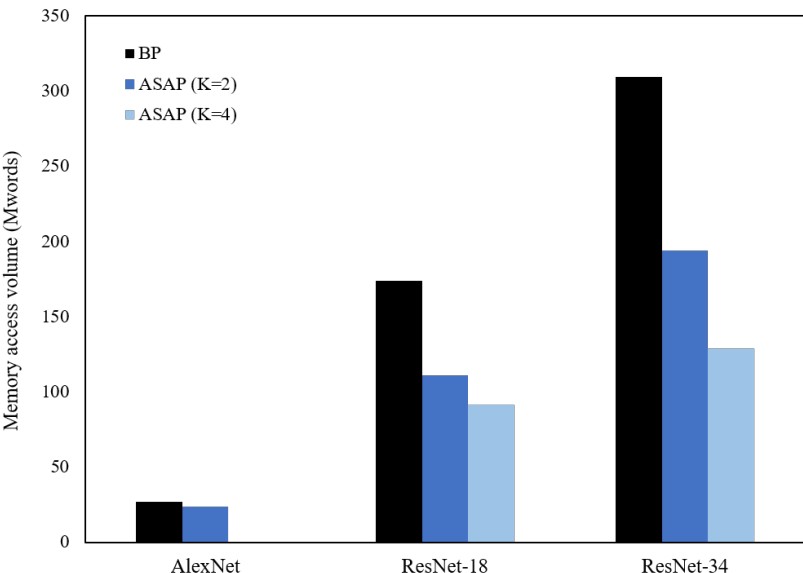

Figure 11: External memory access comparisons.

In the same way as Mostafa et al. [46] did, we estimate the memory access count when implementing our algorithm in hardware, assuming that the compute device has sufficient on-chip memory to buffer

the output activations. When processing the forward pass, the parameters are read from the external memory, and the activations calculated by the compute device are written back to the external memory. Hence, the numbers of read and write accesses are

$$(n_{read},\, n_{write}) = (\boldsymbol{P},\, \boldsymbol{A}) \tag{48}$$

where $\boldsymbol{P}$ and $\boldsymbol{A}$ denote the number of parameters and activations. In the backward pass, the parameters and activations are read from the memory and and the calculated weight changes are written back to the memory. The numbers of read and write accesses are

$$(n_{read}, n_{write}) = (\boldsymbol{P} + \boldsymbol{A}, \boldsymbol{P}) \tag{49}$$

Using equations (48) and (49), we calculate the number of external memory accesses in AlexNet, ResNet-18, and ResNet-34 for backpropagation and ASAP with block sizes of 2 and 4. Note that we only experimented with the block size of 2 in AlexNet since it only has three convolutional layers. Fig. 11 compares the number of external memory accesses of different learning algorithms. Since the ASAP algorithm only needs output activations of some layers to update weights later, we could reduce the amount of activation to be stored in the external memory. As a result, the memory access is reduced in all target networks as shown in Fig. 11. In AlexNet, the number of weight parameters is significantly larger than the number of activations, so the effect of activation sharing is reduced. However, when the network has more convolutional layers as in ResNet-18 and ResNet-34, external memory access can be greatly reduced by using ASAP. Furthermore, this effect is amplified by increasing the block size since more layers would share the same activation. Therefore, the ASAP algorithm is a good candidate for efficient hardware implementation.

# G Pseudocode

**Algorithm 1** Activation Sharing with Asymmetric Paths

1: **procedure** FORWARD PROPAGATION                                  ▷ Determine shared activation
2:      $\tilde{\boldsymbol{h}}_1 = \boldsymbol{h}_1$
3:      $\boldsymbol{h}_{1,0} = \phi(\boldsymbol{W}_{1,0}\boldsymbol{h}_1 + \boldsymbol{b}_{1,0})$
4:      **for** $m = 1$ to $M$ **do**
5:          **for** $k = 1$ to $K$ **do**
6:              $\boldsymbol{h}_{m,k} = \phi(\boldsymbol{W}_{m,k}\boldsymbol{h}_{m,k-1} + \boldsymbol{b}_{m,k})$
7:          **end for**
8:          $\tilde{\boldsymbol{h}}_{m+1} = \boldsymbol{h}_{m,K-1}$
9:          $\boldsymbol{h}_{m+1,0} = \boldsymbol{h}_{m,K}$
10:     **end for**
11: **end procedure**
12:
13: **procedure** BACKWARD PROPAGATION              ▷ Propagate error through feedback weights
14:     $\boldsymbol{y}_{M,K} = softmax(\boldsymbol{h}_{M,K}), \boldsymbol{\delta}_{M,K} = \boldsymbol{y}_{M,K} - \boldsymbol{y}$
15:     **for** $m = M$ to $1$ **do**
16:         **for** $k = K$ to $1$ **do**
17:             $\boldsymbol{\delta}_{m,k-1} = \phi' \boldsymbol{B}_{m,k}^T \boldsymbol{\delta}_{m,k}$
18:         **end for**
19:     **end for**
20: **end procedure**
21:
22: **procedure** WEIGHT UPDATE                          ▷ Use shared activation for weight changes
23:     **for** $m = 1$ to $M$ **do**
24:         **for** $k = 1$ to $K$ **do**
25:             $\Delta\boldsymbol{W}_{m,k} = \boldsymbol{\delta}_{m,k}\tilde{\boldsymbol{h}}_m^T - \lambda\boldsymbol{W}_{m,k}$
26:             $\Delta\boldsymbol{B}_{m,k} = \boldsymbol{\delta}_{m,k}\tilde{\boldsymbol{h}}_m^T - \lambda\boldsymbol{B}_{m,k}$
27:         **end for**
28:     **end for**
29: **end procedure**