# OpenReview forum: "Activation Sharing with Asymmetric Paths Solves Weight Transport Problem without Bidirectional Connection"
_NeurIPS.cc/2021/Conference — NeurIPS 2021 Poster_

### Official Review · Reviewer_i45S · 2021-07-07

**Rating:** 7
**Confidence:** 3

**Summary:**

Methods like backpropagation require the updates to be propagated along the same pathways and weights as the forward computation step (i.e. the "weight transport problem"). To solve this problem, the authors propose activation sharing, which uses random fixed weights both to forwardpropagate activations and to backpropagate error, without bidirectional connections. It is argued that this is more biologically plausible, and can be used to train deep networks (which previous methods could not do).


**Limitations And Societal Impact:**

The biggest real-world limitation is that the method does not perform as well as backprop. This is unfortunate, but also understandable. The authors do mention that ASAP has a lower memory footprint, but there are also other methods that can reduce memory footprints of neural networks trained using backprop.
Given that this method is worse than backprop, and it is also not easy to implement, I cannot see any practical use for it.

On the theoretical side, although the ideas here are interesting, I take issue with the term "biologically plausible" and the appeal to biological networks. Given that cognitive neuroscience has not yet proceeded to a point where we understand how patterns and learning occur in brains, it is extremely premature to try and train networks that match biological neurons on the surface, and claim that we can expect better performance because they are biology inspired. To say that these networks behave more similarly to biological neurons is true only on the surface, and the claim that these networks should therefore be better or superior (in any metric, not just predictive performance) is completely unfounded (and in fact, we can see that the more "inspiration" we draw from biological neurons, the worse our artificial networks tend to be). In this particular case, humans can perform image classification nearly perfectly, and better than the best CNNs trained with backprop. And these CNNs trained with backprop do better than any networks trained using other methods (including ASAP). To clarify, I do not blame (or penalize) the authors for appealing to biological networks, since I think this is a bigger issue in the theoretical ML community as a whole, but I do implore them to soften the language and recognize the severe limitations that prevent us from claiming homology between artificial neural networks and biological neural networks (at least in this decade). I encourage the authors to explicitly clarify that: 1) biological neurons are not yet understood, so drawing inspiration from the little we know does not improve our chances at building better artificial networks; 2) the artificial networks trained using ASAP (and similar methods) do not improve our understanding of biological neurons at all; and 3) the artificial networks trained using ASAP (and similar methods) do not necessarily resemble biological networks (other than the weight transport problem, which is of arguable importance) more than other techniques like backprop. Again, I do not hold the authors accountable for this, and this does not affect the review I gave.

**Main Review:**

The ideas presented in this manuscript are very interesting, and it is nice to see that one can get reasonably good performance in deep neural networks even while propagating through random weights (i.e. the network learns how to learn). This paper is written well, in that it explains the significance and relevance of existing methods, as well as their limitations. The benchmarks were also selected well and were clear. In order to improve the clarity of the mathematical definitions of the ASAP algorithm, it would be preferable to redefine metrics like $\delta$ and $h$ occasionally, as it is easy to forget what they mean given the large number of different symbols being used.


**Time Spent Reviewing:**

3

---

> ### Author Response · Authors · 2021-08-10
> **Reference to Reviewer i45S**
>
> [1] Timothy P Lillicrap, Daniel Cownden, Douglas B Tweed, and Colin J Akerman. Random synaptic feedback weights support error backpropagation for deep learning. Nature communications, 2016.
>
> [2] Joao Sacramento and Rui Ponte Costa and Yoshua Bengio and Walter Senn (2018). Dendritic cortical microcircuits approximate the backpropagation algorithm. NeurIPS 2018, December 3-8, 2018, Montréal, Canada (pp. 8735–8746).
>
> [3] Lillicrap, T. P., Santoro, A., Marris, L., Akerman, C. J., & Hinton, G. Backpropagation and the brain. Nature Reviews Neuroscience, 21(6), 335-346, 2020.
>
> [4] Hesham Mostafa and Vishwajith Ramesh and Gert Cauwenberghs. Deep supervised learning using local errors. CoRR, abs/1711.06756. 2017.
>
> [5] Arild Nokland and Lars Hiller Eidnes (2019). Training Neural Networks with Local Error Signals. ICML 2019, 9-15 June 2019, Long Beach, California, USA (pp. 4839–4850). PMLR.
>
> [6] Roman Pogodin and Peter E. Latham (2020). Kernelized information bottleneck leads to biologically plausible 3-factor Hebbian
> learning in deep networks. NeurIPS 2020, December 6-12, 2020, virtual.
>
> [7] Bernd Illing and Wulfram Gerstner and Guillaume Bellec. Towards truly local gradients with CLAPP: Contrastive, Local And Predictive Plasticity. CoRR, abs/2010.08262. 2020.

---

> ### Author Response · Authors · 2021-08-10
> **Response to Reviewer i45S**
>
> We thank the reviewer for their time in carefully reviewing our submission and providing valuable comments. Please find our responses below:
> \
> \
> **Q1)** In order to improve the clarity of the mathematical definitions of the ASAP algorithm, it would be preferable to redefine metrics like $\delta$ and $h$ occasionally, as it is easy to forget what they mean given the large number of different symbols being used.
> \
> \
> **A1)** We thank the reviewer for this helpful comment. In the revised paper, we will redefine metrics in several places for better readability. In addition, we will add a table summarizing all metrics in the final version.
> \
> \
> \
> \
> **Q2)** I encourage the authors to explicitly clarify that: 1) biological neurons are not yet understood, so drawing inspiration from the little we know does not improve our chances at building better artificial networks; 2) the artificial networks trained using ASAP (and similar methods) do not improve our understanding of biological neurons at all; and 3) the artificial networks trained using ASAP (and similar methods) do not necessarily resemble biological networks (other than the weight transport problem, which is of arguable importance) more than other techniques like backprop.
> \
> \
> **A2)** We appreciate the reviewer’s insightful comments. We agree that our current understanding of biological neural networks is based on just a handful amount of evidence, and what we know could be entirely changed as we get to know better how the brain works in the future.
>
> &nbsp;&nbsp;&nbsp;&nbsp;Nevertheless, the study of brain-inspired algorithms could benefit both cognitive science and machine learning communities. For example, although the Feedback Alignment [1] algorithm itself cannot be asserted to be more biologically plausible than backpropagation, researchers have found their use in predicting and explaining biological models [2, 3]. In addition, a brain-inspired local learning scheme sparked the development of hardware-efficient learning algorithms with similar performance to backpropagation [4, 5, 6, 7]. As suggested by the reviewer, we will soften the language and discuss the raised issues clearly in the camera-ready version if accepted.

---

### Official Review · Reviewer_52YK · 2021-07-16

**Rating:** 4
**Confidence:** 4

**Summary:**

This paper proposes a new learning rule in which the activations of earlier layers (potentially earlier than the presynaptic layer) are used in weight updates -- that is, activations are "shared" for use in weight updates of multiple subsequent layers.  This causes weights to partially align with the identity matrix through a feedback alignment-like mechanism, allowing useful updates to be performed without bidirectional connections between feedforward / feedback pathways (as is needed in algorithms like Kolen-Pollack / weight mirror)

**Limitations And Societal Impact:**

The authors do address limitations, and I don't think that addressing potential negative societal impact is necessary given the subject of this paper.

**Main Review:**

I find the forward alignment idea, and its application (setting C = I) to the activation sharing algorithm interesting.  It is neat that this approach can work reasonably well on tasks like CIFAR.  However, I'm not convinced by the paper's motivation.  Sure, direct bidirectional connections between feedforward / feedback may not be common enough to exactly implement the Kolen-Pollack / weight mirror algorithms, but effective bidirectional connectivity via multisynaptic pathways should do the trick.  This paper's approach induces a lot of extra complexity -- how, mechanistically, are activations shared from earlier layers?  Is there any reason to think this is more biologically plausible than effective bidirectional connectivity?  Moreover, this complexity is introduced at a performance cost -- ASAP clearly lags behind backprop, while weight mirror / kollen-pollack have been shown to be very competitive with backprop (why are they not compared against directly?  Also, see https://arxiv.org/abs/2003.01513, which should probably also be cited / compared to).

Overall, I am not convinced that this paper makes practical ML advances or moves closer to biological plausibility, and hence I don't think it's ready for acceptance.

**Time Spent Reviewing:**

2

---

> ### Author Response · Authors · 2021-08-10
> **Reference to Reviewer 52YK**
>
> [1] Mohamed Akrout, Collin Wilson, Peter Humphreys, Timothy Lillicrap, and Douglas B Tweed. Deep learning without weight transport. In Advances in Neural Information Processing Systems, volume 32, 2019.
>
> [2] Andrey Gushchin and Ao Tang. Total wiring length minimization of c. elegans neural network: a constrained optimization approach. PloS one, 2015.
>
> [3] Marion Langen, Egemen Agi, Dylan J Altschuler, Lani F Wu, Steven J Altschuler, and Peter Robin
> Hiesinger. The developmental rules of neural superposition in drosophila. Cell, 2015.
>
> [4] Sanford L Palay and Victoria Chan-Palay. Cerebellar cortex: cytology and organization. 2012.
>
> [5] Joao Sacramento and Rui Ponte Costa and Yoshua Bengio and Walter Senn. Dendritic cortical microcircuits approximate the backpropagation algorithm. NeurIPS 2018, December 3-8, 2018, Montréal, Canada (pp. 8735–8746).
>
> [6] György Buzsáki and Kenji Mizuseki. The log-dynamic brain: how skewed distributions affect network operations. In Nature Reviews Neuroscience, pages 264—-278. 2014
>
> [7] David Fitzpatrick. The Functional Organization of Local Circuits in Visual Cortex: Insights from the Study of Tree Shrew Striate Cortex. Cerebral Cortex, 1996.
>
> [8] Seung Wook Oh, Julie A. Harris, and Hongkui Zeng. A mesoscale connectome of the mouse brain. In Nature, pages 207—-214. 2014.
>
> [9] Alex M. Thomson. Neocortical layer 6, a review. Front. Neuroanat, 2010.
>
> [10] Sato, T. K. Long-range connections enrich cortical computations. Neuroscience Research, 162, 1-12, 2021.
>
> [11] Kathleen S Rockland and Agnes Virga. Terminal arbors of individual “feedback” axons projecting from area v2 to v1 in the macaque monkey: a study using immunohistochemistry of anterogradely transported phaseolus vulgaris-leucoagglutinin. Journal of Comparative 392 Neurology, 1989.
>
> [12] Penelope C Murphy and Adam M Sillito. Functional morphology of the feedback pathway from area 17 of the cat visual cortex to the lateral geniculate nucleus. Journal of Neuroscience, 1996.
>
> [13] Rockland, K. S., Kaas, J. H., & Peters, A. (Eds.). Cerebral Cortex: Volume 12: Extrastriate Cortex in Primates (Vol. 12). Springer Science & Business Media, 2013.
>
> [14] Rockland, K. S., & Van Hoesen, G. W. Direct temporal-occipital feedback connections to striate cortex (V1) in the macaque monkey. Cerebral cortex, 4(3), 300-313. 1994.
>
> [15] Rockland, K. S., & Drash, G. W. Collateralized divergent feedback connections that target multiple cortical areas. Journal of Comparative Neurology, 373(4), 529-548. 1996

---

> ### Author Response · Authors · 2021-08-10
> **Response to Reviewer 52YK**
>
> We thank the reviewer for their time in carefully reviewing our submission and providing valuable comments. Please find our responses below:
> \
> \
> **Q1)** This paper's approach induces a lot of extra complexity -- how, mechanistically, are activations shared from earlier layers? Is there any reason to think this is more biologically plausible than effective bidirectional connectivity?
> \
> \
> **A1)** We thank the reviewer for raising a very important issue. We apologize that our original manuscript did not discuss this in detail. Akrout et al. [1], who proposed algorithms using the bidirectional connections, claim that one-to-one paring that biologically supports a bidirectional connection is observed in some organisms [2, 3, 4]. Although we agree that bidirectional connections may exist in the biological neural network, there exist structural limitations derived from bidirectional connections, as discussed in prior works [2, 5]. In fact, Akrout et al. [1] also acknowledge this limitation and mention that “something less than strict one-to-one wiring may suffice for effective learning, and may itself be learned”.
>
> &nbsp;&nbsp;&nbsp;&nbsp;Our ASAP algorithm could significantly relax these structural constraints. ASAP does not require bidirectional connections, and activation sharing can be implemented using a one-way skip connection (i.e., a connection between non-adjacent layers). Those one-way skip connections are frequently found in living organisms [6, 7, 8, 9]. In addition, it has recently been confirmed that long-range connections play an important role in cortical computation [10]. The observations in the visual cortex also show that reciprocal connections are generally spatially asymmetric, consisting of multi-step one-way connections [11, 12].
>
> &nbsp;&nbsp;&nbsp;&nbsp;Of course, our algorithm could also be more spatially complex. However, this may make more sense than a simple and symmetrical bidirectional connection based on observations from the visual cortex. The visual cortex consists of fairly complex and asymmetrical structures. For example, in the visual cortex, feedforward terminations in layer 4 cannot make direct contact with feedback neurons in layer 3A, but only with the feedback neurons of layer 6 along the distal extent of their dendrites [13]. From these observations, Rockland et al. [13] claim that feedforward termination probably uses multiple short- and long-chain and interlaminar routes to connect to feedback neurons for any two interconnected areas. Also, reciprocal connections are generally spatially asymmetric, consisting of multi-step one-way connections. This is supported by the observation that terminal arbors of feedforward connections are relatively concentrated, whereas those of feedback connections are generally divergent [11, 12]. Third, many unidirectional connections are observed in the feedback direction. In area TEO, the neurons receive input from area 36 and possibly area 35, but it does not project itself into these perirhinal regions [14, 15].
>
> &nbsp;&nbsp;&nbsp;&nbsp;The reviewer is certainly correct that a bidirectional connection could be implemented using a multi-synaptic pathway. Therefore, we agree with the reviewer’s comment that one-to-one paring might be biologically plausible. However, our ASAP algorithm may coincide well with such complex biological systems consisting of asymmetric one-way skip connections. These points will be later reflected in the camera-ready version.
> \
> \
> \
> \
> **Q2)** Moreover, this complexity is introduced at a performance cost -- ASAP clearly lags behind backprop, while weight mirror / kollen-pollack have been shown to be very competitive with backprop (why are they not compared against directly? Also, see https://arxiv.org/abs/2003.01513, which should probably also be cited / compared to).
> \
> \
> **A2)** The reviewer is certainly correct that our algorithm introduces additional spatial complexity as well as performance degradation, whereas KP and WM algorithms using bidirectional connections achieve similar performance to backpropagation. In the original manuscript, we did not directly compare ASAP with these algorithms because our main goal was to develop an algorithm that could match complex biological systems by overcoming the structural limitations derived from bidirectional connections. However, we will include comparisons with those algorithms and discuss the limitations of our algorithm in the updated manuscript.
>
> &nbsp;&nbsp;&nbsp;&nbsp;As discussed in our response to Q1, we believe that our ASAP algorithm could better represent complex biological systems consisting of asymmetric one-way skip connections, at the expense of some amount of performance degradation. Therefore, we might break the trade-off between biological plausibility and training performance by taking advantage of both algorithms. In other words, if bidirectional connections are made between some feedforward and feedback neurons, while the rest are implemented using one-way skip connections as in ASAP, performance can be further improved while still supporting complex biological models.
>
> &nbsp;&nbsp;&nbsp;&nbsp;As suggested by the reviewer, we will add performance comparisons with the Weight-Mirror (WM) and Kolen-Pollack (KP) algorithms, and further details our algorithm’s limitations. Furthermore, we will discuss that our ASAP algorithm and the WM and KP algorithms could complement each other to support complex biological neural networks while achieving high performance.

---

### Official Review · Reviewer_aqbm · 2021-07-16

**Rating:** 7
**Confidence:** 3

**Summary:**

The paper proposes a novel approximation to backpropagation that solves the need for bidirectional connections between the forward pass neurons, and the backwards pass circuitry. This is done via activation sharing, meaning that the weight update is computed by the backward pass circuitry and activations of some layer prior to the current one.

**Limitations And Societal Impact:**

The authors have adequately addressed the limitations and potential negative societal impact of their work.

**Main Review:**

Originality: The activation sharing idea is new, and the overall algorithm (ASAP) combines it with the Kolen-Pollack algorithm to also avoid the weight transport problem. However, the paper might benefit from comparison (in terms of both biological plausibility and performance) of their algorithm with other biologically plausible learning rules that do not approximate backprop, but still achieve competitive performance. Some examples include layer-wise learning rules [1-4] and equilibrium propagation [5]. It might also be a good idea to discuss the predictive coding view on backprop approximations (e.g. [6]).

Quality: The theoretical results in the paper are sound, and the experiments good enough (although again, should be compared with alternative methods mentioned above).

Clarity: Overall, the paper is well-written. However, I think a few claims should be corrected.
In the abstract, “also achieving good performance even on deep convolutional neural networks, unlike other biologically plausible algorithms” does not hold as alternative methods (see citations above) have achieved comparable performance too.
In line 26, “it is impossible to make their weights identical without explicitly passing the weights between the paths” contradicts the weight mirror and Kolen-Pollack algorithms discussed later, as they do exactly that. Also, I would change dots over variables to tilde, as dots usually denote a time derivative.

Significance: The paper proposes an interesting approximation to backprop. However, I have a major concern about the biological plausibility of the proposed algorithm.
First, the bidirectional connections might actually be a good idea, for instance if the backward pass is done within the neuron via apical dendrites [7].
Second, activation sharing makes the final algorithm more biologically implausible that just Kolen-Pollack, in my opinion. In approximations to backprop, the error signal is neuron-specific: each neuron receives its own $\\delta$, and uses the pre-synaptic activity to do the weight update. In ASAP, each synapse, not neuron, receives a unique error signal that doesn’t use any information available in either pre- or post-synaptic neurons. The authors should discuss how this can be implemented.

[1] Deep supervised learning using local errors, Hesham Mostafa, Vishwajith Ramesh, and Gert Cauwenberghs

[2] Training neural networks with local error signals, Arild Nøkland and Lars Hiller Eidnes

[3] Kernelized information bottleneck leads to biologically plausible 3-factor Hebbian learning in deep networks, Roman Pogodin and Peter Latham

[4] Local plasticity rules can learn deep representations using self-supervised contrastive predictions, Bernd Illing, Jean Ventura, Guillaume Bellec, Wulfram Gerstner

[5] Scaling Equilibrium Propagation to Deep ConvNets by Drastically Reducing Its Gradient Estimator Bias, Axel Laborieux, Maxence Ernoult, Benjamin Scellier, Yoshua Bengio, Julie Grollier and Damien Querlioz

[6] Predictive Coding Approximates Backprop along Arbitrary Computation Graphs, Beren Millidge, Alexander Tschantz, Christopher L. Buckley

[7] Dendritic cortical microcircuits approximate the backpropagation algorithm, João Sacramento, Rui Ponte Costa, Yoshua Bengio, Walter Senn

**Update: increased the score from 6 to 7 (explanation in the thread below)**


**Time Spent Reviewing:**

3

---

> ### Author Response · Authors · 2021-08-10
> **Reference to Reviewer aqbm**
>
> [1] Hesham Mostafa and Vishwajith Ramesh and Gert Cauwenbergh. Deep supervised learning using local errors. CoRR, abs/1711.06756, 2017.
>
> [2] Arild Nokland and Lars Hiller Eidnes. Training Neural Networks with Local Error Signals. ICML 2019, 9-15 Long Beach, California, USA (pp. 4839–4850). PMLR.
>
> [3] Roman Pogodin and Peter E. Latham. Kernelized information bottleneck leads to biologically plausible 3-factor Hebbian learning in deep networks. NeurIPS 2020, December 6-12, 2020, virtual.
>
> [4] Bernd Illing and Wulfram Gerstner and Guillaume Bellec. Towards truly local gradients with CLAPP: Contrastive, Local And Predictive Plasticity. CoRR, abs/2010.08262, 2020.
>
> [5] Laborieux, A., Ernoult, M., Scellier, B., Bengio, Y., Grollier, J., & Querlioz, D. Scaling equilibrium propagation to deep convnets by drastically reducing its gradient estimator bias. Frontiers in neuroscience, 15, 129, 2021.
>
> [6] Beren Millidge and Alexander Tschantz and Christopher L. Buckley. Predictive Coding Approximates Backprop along Arbitrary Computation Graphs. CoRR, abs/2006.04182, 2020.
>
> [7] Mohamed Akrout, Collin Wilson, Peter Humphreys, Timothy Lillicrap, and Douglas B Tweed. Deep learning without weight transport. In Advances in Neural Information Processing Systems, volume 32, 2019.
>
> [8] Andrey Gushchin and Ao Tang. Total wiring length minimization of c. elegans neural network: a constrained optimization approach. PloS one, 2015.
>
> [9] Marion Langen, Egemen Agi, Dylan J Altschuler, Lani F Wu, Steven J Altschuler, and Peter Robin Hiesinger. The developmental rules of neural superposition in drosophila. Cell, 2015.
>
> [10] Sanford L Palay and Victoria Chan-Palay. Cerebellar cortex: cytology and organization. 2012.
>
> [11] Joao Sacramento and Rui Ponte Costa and Yoshua Bengio and Walter Senn. Dendritic cortical microcircuits approximate the backpropagation algorithm. NeurIPS 2018, December 3-8, 2018, Montréal, Canada (pp. 8735–8746), 2018.
>
> [12] György Buzsáki and Kenji Mizuseki. The log-dynamic brain: how skewed distributions affect network operations. In Nature Reviews Neuroscience, pages 264—-278. 2014
>
> [13] David Fitzpatrick. The Functional Organization of Local Circuits in Visual Cortex: Insights from the Study of Tree Shrew Striate Cortex. Cerebral Cortex, 1996.
>
> [14] Seung Wook Oh, Julie A. Harris, and Hongkui Zeng. A mesoscale connectome of the mouse brain. In Nature, pages 207—-214. 2014.
>
> [15] Alex M. Thomson. Neocortical layer 6, a review. Front. Neuroanat, 2010.
>
> [16] Sato, T. K. Long-range connections enrich cortical computations. Neuroscience Research, 162, 1-12, 2021.
>
> [17] Kathleen S Rockland and Agnes Virga. Terminal arbors of individual “feedback” axons projecting from area v2 to v1 in the macaque monkey: a study using immunohistochemistry of anterogradely transported phaseolus vulgaris-leucoagglutinin. Journal of Comparative 392 Neurology, 1989.
>
> [18] Penelope C Murphy and Adam M Sillito. Functional morphology of the feedback pathway from area 17 of the cat visual cortex to the lateral geniculate nucleus. Journal of Neuroscience, 1996.

---

> ### Author Response · Authors · 2021-08-10
> **Response to Reviewer aqbm**
>
> We thank the reviewer for their time in carefully reviewing our submission and providing valuable comments. Please find our responses below:
> \
> \
> **Q1)** The paper might benefit from comparison (in terms of both biological plausibility and performance) of their algorithm with other biologically plausible learning rules that do not approximate backprop, but still achieve competitive performance.
> \
> \
> **A1)** Since the purpose of our paper is to solve the weight transport problem without a bidirectional connection, feedback alignment and direct feedback alignment were chosen for comparisons in the original manuscript. However, as the reviewer recommended, we compared ASAP with other algorithms based on local learning [1, 2, 3]. Those algorithms are applied to training ResNet-34 on the CIFAR-10 dataset, where we used identical hyperparameters, but different learning rates are used for optimal results. Additional hyperparameters for implementing the local loss follow the values reported in their paper. The experimental results are as follows:
> \
> \
> &nbsp;&nbsp;&nbsp;&nbsp;&nbsp;&nbsp;BP: 95.18%
>
> &nbsp;&nbsp;&nbsp;&nbsp;&nbsp;&nbsp;Using local errors [1]: 69.8%
>
> &nbsp;&nbsp;&nbsp;&nbsp;&nbsp;&nbsp;With local error signals [2]: 87.57%
>
> &nbsp;&nbsp;&nbsp;&nbsp;&nbsp;&nbsp;Kernelized information bottleneck [3]: 77.2%
>
> &nbsp;&nbsp;&nbsp;&nbsp;&nbsp;&nbsp;ASAP (ours): 93.97%
> \
> \
> &nbsp;&nbsp;&nbsp;&nbsp;Although prior works report excellent training performance on VGG-like models, being very close to that of BP, they exhibit noticeable performance degradation when applied to ResNet-34, which was also reported in [2]. We speculate that the skip connections in ResNet models do not improve training performance for local learning as the errors do not backpropagate through layers. On the other hand, our ASAP algorithm takes advantage of shortcut connections, as discussed in Section 4 and Appendix. In terms of biological plausibility, the algorithms presented in [1] and [2] still suffer from weight transport problem in auxiliary layers. The algorithm in [3] solves the weight transport problem without using bidirectional connections, but it exhibits much lower training performance as shown above. Nevertheless, local learning algorithms are meaningful in that it shows that layer-wise learning is possible. In conclusion, while local learning enables layer-wise learning, ASAP solves the weight transport problem without the structural constraints of bidirectional connection while achieving significantly better training performance.
>
> &nbsp;&nbsp;&nbsp;&nbsp;Note that the Contrastive, Local And Predictive Plasticity Algorithm (CLAPP) [4] is designed for self-supervised learning, and hence the direct comparison is not feasible. The Equilibrium Propagation algorithm shows the competitive performance when used for the image classification task [5]. However, the algorithm needs to be heavily modified to train our target models (e.g., ResNet-34), and we would need extra time for this experiment. We will update the experimental results in the final version. In the future study, from the perspective of predictive coding, we plan to directly implement our ASAP as predictive coding or transform the computation graph of the predictive coding into an ASAP-like way.
> \
> \
> \
> \
> **Q2)** The claim that “also achieving good performance even on deep convolutional neural networks, unlike other biologically plausible algorithms” does not hold because there are other algorithms with comparable performance.
> \
> \
> **A2)** We thank the reviewer for bringing this up, and we agree that the sentence may be misleading. Other biologically plausible algorithms [1, 2, 3, 4, 5] could lead to competitive results in deep convolutional networks. Therefore, this expression will be modified to "also achieving good performance even on deep convolutional neural networks." Other similar claims in the paper will also be modified.
> \
> \
> \
> \
> **Q3)** The claim that “it is impossible to make their weights identical without explicitly passing the weights between the paths” contradicts the weight mirror and Kolen-Pollack algorithms.
> \
> \
> **A3)** We apologize for our mistake in the original manuscript. As pointed out by the reviewer, even if the initial values of the forward weight and the feedback weight are different, the two weights could become identical if those algorithms are employed. We will remove this sentence in the updated version.
> \
> \
> \
> \
> **Q4)** Also, I would change dots over variables to tilde, as dots usually denote a time derivative.
> \
> \
> **A4)** The dots will be changed to tilde in the updated paper.
> \
> \
> \
> \
> **Q5)** The bidirectional connections might actually be a good idea, for instance if the backward pass is done within the neuron via apical dendrites.
> \
> \
> **A5)** We thank the reviewer for raising this issue. We apologize that our original manuscript did not discuss this point in detail. Akrout et al. [7], who proposed algorithms using the bidirectional connections, claim that one-to-one paring that biologically supports a bidirectional connection is observed in some organisms [8, 9, 10]. Although we agree that bidirectional connections may exist in the biological neural network, there exist structural limitations derived from bidirectional connections, as discussed in other prior works [8, 11]. In fact, Akrout et al. [7] also acknowledge this limitation and mention that “something less than strict one-to-one wiring may suffice for effective learning, and may itself be learned”.
>
> &nbsp;&nbsp;&nbsp;&nbsp;Our ASAP algorithm could significantly relax these structural constraints. ASAP does not require bidirectional connections, and activation sharing can be implemented using a one-way skip connection (i.e., a connection between non-adjacent layers). Those one-way skip connections are frequently found in living organisms [12, 13, 14, 15]. In addition, it has recently been confirmed that long-range connections play an important role in cortical computation [16]. The observations in the visual cortex also show that reciprocal connections are generally spatially asymmetric, consisting of multi-step one-way connections [17, 18].
>
> &nbsp;&nbsp;&nbsp;&nbsp;As pointed out by the reviewer, using multi-compartment neurons and apical dendrite activity for error propagation could also solve this issue [11]. However, this approach was only tested on a simple 4-layer MLP network and MNIST dataset, and more experiments are required to verify if their approach could scale to deep convolutional networks on a large dataset. Nevertheless, we agree with the reviewer’s comment that one-to-one paring might actually be biologically plausible, so we will soften the language and add this discussion in the camera-ready version if accepted.
> \
> \
> \
> \
> **Q6)** In ASAP, each synapse, not neuron, receives a unique error signal that doesn’t use any information available in either pre- or post-synaptic neurons. The authors should discuss how this can be implemented.
> \
> \
> **A6)** This is a very valid point, and we thank the reviewer for bringing this issue to our attention. In our algorithm, we agree that each synapse should receive a unique error signal, contrary to simply using pre- or post-synaptic activities, for an update. However, this issue can be addressed if the pre-synaptic neurons could be modulated in a way that they mimic the neurons in the previous layers. In other words, in Figure 3b, if the feedforward neurons in layer $l+1$ generate activities identical to the shared activation $h_{l}$ then each synapse could be updated using information from pre- and post-synaptic neurons as usual.
> This problem was also discussed in prior work for the Weight Mirror algorithm (WM)[7] because the algorithm does not use pre- and post-synaptic neuron information to update the feedback weights. The weight update in the Weight Mirror algorithm follows the equation below:
> \
> \
> &nbsp;&nbsp;&nbsp;&nbsp;&nbsp;&nbsp;$\delta B_{l+1} = h_{l+1} h_{l}$
> \
> \
> &nbsp;&nbsp;&nbsp;&nbsp;Weight Mirror does not use pre-synaptic neuron information $\delta_{l+1}$ and post-synaptic neurons information $\delta_{l}$ to learn the feedback weight $B_{l+1}$, but rather use the signals $h_{l+1}$ and $h_{l}$ from the forward path.
>
> &nbsp;&nbsp;&nbsp;&nbsp;To solve this problem, Akrout et al. [7] use a 'mirror mode' where feedforward neurons adjust the feedback neurons so that they mimic the forward neurons. Therefore, due to the strong projection from feedforward neurons in the mirror mode, the feedback neuron activities $\delta_{l}$ mimics $h_{l}$, making it possible to update weights only using pre- and post-synaptic neuron information.
>
> &nbsp;&nbsp;&nbsp;&nbsp;In our algorithm, the problem raised by the reviewer could be solved in a similar way:
> \
> \
> \
> (1) Before propagating the error, each neuron mimics shared activation through projection through the skip connection in Fig. 3b in mirror mode. That is, neurons of layers $l+1$ and $l+2$ mimic the neurons in layer $l$ neurons which are providing shared activation $h_{l}$.
>
> (2) Forward-feedback weights are learned using information from pre- and post-synaptic neurons while propagating the errors backward.
> \
> \
> \
> &nbsp;&nbsp;&nbsp;&nbsp;Akrout et al. [7] point out that the Weight Mirror algorithm has a disadvantage that it needs to alternately perform weight mirror mode and engaged mode, where updates forward weights by sensory input $\delta$. This is because, if mirror mode is performed for all neurons at once without engaged mode, projection occurs from forward neurons to feedback neurons at once, and activation information for forward weight updates disappears from forward neurons. However, our ASAP algorithm is free from this problem because the projection occurs between feedforward neurons using skip connections. This point will be included in the final manuscript along with a detailed figure.

---

> > ### Comment · Reviewer_aqbm · 2021-08-18
> > **Response (more experiments if possible)**
> >
> > Thank you for the response! I’m mostly satisfied with your replies. But I’d like to add a few more comments, and continue the conversation.
> >
> > > plausibility and synapse-specific error signals
> >
> > I’d say the training scheme you propose in A6 is quite complicated, and doesn’t make the algorithm any more plausible than backprop/weight mirror/etc. I agree that Akrout et.al. faced a similar issue, but that brings us back to Q5 and A5: if we keep the errors in the same neuron (i.e. the backwards pass goes through the forward neurons), the pre- and post-synaptic activity is at least available at each synapse.
> >
> > I think another way to frame my concern is that while you do break bidirectionality, you sort of still need it to bring activities from the earlier layers to both the forward and backward neurons.
> >
> > However, I think there’s a simple fix that would make the result much stronger: actually break bidirectionality at the cost of non-mirrored weights. You can use $h_l$ to update the forward weights, $\Delta W_{l+1}\propto - \delta_{l+1} h_l$, but $h_{l-2}$ to update the backward weights: $B_{l+1}\propto - \delta_{l+1} h_{l-2}$. This scheme would probably result in a performance hit, but in my opinion would solve the bidirectional connections problem more naturally.
> >
> > It’d be great to run at least preliminary experiments with that scheme until the end of the discussion period, although I understand that there’s not much time left.
> >
> > > Q1/A1
> >
> > These results are quite interesting, and I guess it’s not surprising that layer-wise rules perform much worse on such a deep network. But I also find ResNet-34 a total overkill for CIFAR10. For the final version (not for the discussion!), I think a ResNet-18 (or the CIFAR10 variant from the resnet paper) would fit better. Another option is to have both 18 and 34 variants to show the advantage of ASAP w.r.t. depth.

---

> > > ### Author Response · Authors · 2021-08-25
> > > **Response to additional review**
> > >
> > > We thank the reviewer for providing constructive feedback and continuing discussion that would help improve our submission. Please find our responses below:
> > > \
> > > \
> > > **1)	Implementing forward and feedback paths in a single neuron**
> > > \
> > > \
> > > &nbsp;&nbsp;&nbsp;&nbsp;We agree with the reviewer that the bidirectional connections between the forward and feedback neurons could be biologically implemented using a multi-compartment neuron model with apical dendrites. As pointed out by the reviewer, Sacramento et al. [1] show that even backpropagation could be realized by receiving errors through apical dendrites. However, implementing both forward and feedback paths in the same neuron could lead to some potential issues. First, while this scheme resolves the bidirectional connection between the forward and feedback paths, it necessitates another bidirectional connection between the neurons in the adjacent layers. Whittington and Bogacz [2] point out that this dendritic model (i.e., implementing both paths in a single neuron using a multi-compartment neuron structure) requires one-to-one connections to the next layer, but they argue that there is no evidence that this one-to-one connectivity exists in the neocortex [2]. Mesnard et al. [3] mimic the dendritic model using point neurons by introducing ghost units, and they acknowledge that their model suffers from biological implausibility due to similar inter-layer one-to-one connectivity. Even Sacramento et al. [1], who approximated backpropagation using apical dendrites, also disclose that one-to-one connectivity between layers is a structural constraint in their model architecture.
> > > \
> > > &nbsp;&nbsp;&nbsp;&nbsp;For this reason, we believe that it could be advantageous to implement the forward and feedback paths as individual neurons to remove bidirectional connections between the two paths inside a layer as well as between two adjacent layers. However, the reviewer raised an important issue related to synapse-specific error signals, and we will address this issue in detail below.
> > > \
> > > \
> > > \
> > > \
> > > **2)	Solving synapse-specific error signal issue when forward and backward propagations do not occur within the same neuron**
> > > \
> > > \
> > > &nbsp;&nbsp;&nbsp;&nbsp;If we assume that the forward and feedback paths are implemented separately using individual neurons, our ASAP algorithm can be depicted as Figure 3b in the manuscript. This approach solves the weight transport problem without inter-layer or intra-layer bidirectional connections as discussed above, unlike the backpropagation, WM, and KP algorithms. However, as pointed out by the reviewer, this update rule may require synapse-specific error signals instead of relying on pre- and post-synaptic activities during learning. As we discussed in A6, this problem could be mitigated by using mirror mode [4], in which neurons mimic other neurons during weight updates. For instance, in the mirror mode, the neurons of layers $l+1$ and $l+2$ mimic the neurons in layer $l$ that generate the shared activation $h_{l}$. As a result, the output activations of layers $l+1$ and $l+2$ become $h_{l}$ instead of $h_{l+1}$ and $h_{l+2}$. Then, pre- and post-synaptic activities could be directly used for calculating weight updates, similar to other biologically plausible algorithms.
> > > \
> > > &nbsp;&nbsp;&nbsp;&nbsp; However, we do agree with the reviewer that adding a mirror mode complicates the learning process and hinders its biological plausibility to some extent. Nevertheless, ASAP is still noticeably simpler (and possibly more biologically plausible) than the Weight Mirror (WM) algorithm. In WM, the neural network should alternate between the mirror mode and the engaged mode. The feedback weights are updated in the mirror mode, whereas the forward weights are updated in the engaged mode, because the forward and feedback paths employ different types of activities for weight updates. Contrarily, our ASAP algorithm only needs the mirror mode during error propagation as both paths utilize the same neuronal activities to update weights. One more point to consider is that in the mirror mode, ASAP still requires activity transfer from earlier layers to the forward and backward neurons, as pointed out by the reviewer. This transfer could be realized using a one-way skip connection, which is often found in the brain, such as the visual cortex [5, 6, 7, 8]. This is also consistent with recent observations that long-range skip connections play an important role in cortical computation [9]. The reviewer is certainly correct that providing identical activities to both forward and feedback neurons is another structural constraint of the ASAP algorithm. However, this structure could be constructed only using multiple one-way connections without explicit bidirectional connections, and hence we believe that ASAP achieves a higher degree of biological plausibility than the WM and KP algorithms.
> > > \
> > > \
> > > \
> > > \
> > > **3)	Breaking bidirectionality using non-mirrored weight updates**
> > > \
> > > \
> > > &nbsp;&nbsp;&nbsp;&nbsp;We greatly appreciate the reviewer for suggesting a novel idea that would make our algorithm more biologically plausible, where $h_{l}$ is used to update the forward weights and the shared activation $h_{l-2}$ is used to update the backward weights. We named this learning algorithm Activation Sharing in Feedback Path (ASFP), and the experimental results are as below:
> > >
> > > ||&nbsp;&nbsp;&nbsp;&nbsp;&nbsp;&nbsp;&nbsp;&nbsp;ASAP&nbsp;&nbsp;&nbsp;&nbsp;&nbsp;&nbsp;&nbsp;&nbsp;|&nbsp;&nbsp;&nbsp;&nbsp;&nbsp;&nbsp;&nbsp;&nbsp;ASFP&nbsp;&nbsp;&nbsp;&nbsp;&nbsp;&nbsp;&nbsp;&nbsp;|
> > > |:------:|:---:|:---:|
> > > |AlexNet|78.25|75.62|
> > > |ResNet-18|92.19|61.45|
> > > |ResNet-34|93.97|31.7|
> > > |||||
> > >
> > > &nbsp;&nbsp;&nbsp;&nbsp;Unfortunately, the ASFP algorithm exhibits significant performance degradation or fails to train the network when applied to deep neural networks. We speculate that this performance degradation is due to the difference in the weight update directions of the forward and feedback weights.
> > > However, we further investigated this issue and found a potential solution. We adopted local learning to minimize the difference in the weight update directions in the ASFP algorithm. Specifically, we added a local classifier whenever the number of channels changes in the neural network. The local classifier consists of an adaptive average pooling whose output dimension is 2×2 and a linear layer. The linear layer was trained using Feedback Alignment to maintain biological plausibility. The experimental results are as follows:
> > >
> > > ||&nbsp;&nbsp;&nbsp;&nbsp;&nbsp;&nbsp;&nbsp;&nbsp;ASAP&nbsp;&nbsp;&nbsp;&nbsp;&nbsp;&nbsp;&nbsp;&nbsp;|&nbsp;&nbsp;ASFP with local classifier&nbsp;&nbsp;|&nbsp;&nbsp;using local error signals [10]&nbsp;&nbsp;|
> > > |:------:|:---:|:---:|:---:|
> > > |ResNet-18|92.19|88.46|88.12|
> > > |ResNet-34|93.97|89.28|87.57|
> > > |||||
> > >
> > > &nbsp;&nbsp;&nbsp;&nbsp;Experimental results confirm that ASFP could reliably train deep neural networks with the aid of local learning. We also compared this algorithm with a layer-wise local learning algorithm that demonstrated the best performance in our previous response [10]. ASFP combined with local learning not only outperforms the local learning algorithm but also achieves a higher degree of biological plausibility since the local learning algorithm in [10] uses backpropagation for training local classifiers
> > > \
> > > \
> > > \
> > > \
> > > **4) Conclusion**
> > > \
> > > \
> > > &nbsp;&nbsp;&nbsp;&nbsp;To address the reviewer's questions, we explored the limitations of the dendritic model and showed how our ASAP algorithm could mitigate the synapse-specific error signal issue. Furthermore, thanks to the reviewer's suggestion, we were able to find a competitive algorithm that could achieve high performance even if the forward and backward weights are updated in different directions. These points will be reflected in detail in the final version, along with more experimental results.
> > > \
> > > \
> > > \
> > > [1] Joao Sacramento, Rui Ponte Costa, Yoshua Bengio and Walter Senn. Dendritic cortical microcircuits approximate the backpropagation algorithm. NeurIPS, December 3-8, Montréal, Canada (pp. 8735–8746), 2018.
> > > \
> > > \
> > > [2] Whittington, J. C. and Bogacz, R. Theories of error backpropagation in the brain. Trends in cognitive sciences, 2019.
> > > \
> > > \
> > > [3] Thomas Mesnard, Gaetan Vignoud, Joao Sacramento, Walter Senn and Yoshua Bengio. Ghost Units Yield Biologically Plausible Backprop in Deep Neural Networks. CoRR, 2019.
> > > \
> > > \
> > > [4] Mohamed Akrout, Collin Wilson, Peter Humphreys, Timothy Lillicrap, and Douglas B Tweed. Deep learning without weight transport. In Advances in Neural Information Processing Systems, volume 32, 2019.
> > > \
> > > \
> > > [5] György Buzsáki and Kenji Mizuseki. The log-dynamic brain: how skewed distributions affect network operations. In Nature Reviews Neuroscience, pages 264—-278. 2014
> > > \
> > > \
> > > [6] David Fitzpatrick. The Functional Organization of Local Circuits in Visual Cortex: Insights from the Study of Tree Shrew Striate Cortex. Cerebral Cortex, 1996.
> > > \
> > > \
> > > [7] Seung Wook Oh, Julie A. Harris, and Hongkui Zeng. A mesoscale connectome of the mouse brain. In Nature, pages 207—-214. 2014.
> > > \
> > > \
> > > [8] Alex M. Thomson. Neocortical layer 6, a review. Front. Neuroanat, 2010.
> > > \
> > > \
> > > [9] Sato, T. K. Long-range connections enrich cortical computations. Neuroscience Research, 162, 1-12, 2021.
> > > \
> > > \
> > > [10] Arild Nokland and Lars Hiller Eidnes. Training Neural Networks with Local Error Signals. ICML 2019, 9-15 Long Beach, California, USA (pp. 4839–4850). PMLR.

---

> > > > ### Comment · Reviewer_aqbm · 2021-09-01
> > > > **Score increased to 7**
> > > >
> > > > Thanks for the response!
> > > >
> > > > I still think the synapse-specific errors are a problem, but I think your arguments are valid as well. The new experimental results are quite interesting, and It's great to see that a more relaxed version of the update rule worked too (well, with some aid). I think these experiment should definitely make their way into the final version.
> > > >
> > > >  Overall, I think it's fair to increase the score from 6 to 7 as my concerns have been addressed.

---

### Official Review · Reviewer_Fjkm · 2021-07-17

**Rating:** 6
**Confidence:** 4

**Summary:**

This work presents a new neural network learning algorithm that removes some of the constraints required by the traditional backpropagation algorithm. Constraints such as explicit bidirectional connections and symmetric weights between the forward and feedback paths are not used with the proposed algorithm. Through experiments on multiple popular datasets, the authors demonstrate their algorithm is competitive or better than several other learning algorithms including Feedback Alignment and Direct Feedback Alignment despite having fewer constraints.

This work also provides a thorough overview of the signals propagated through the neural network for algorithms like backpropagation, feedback alignment, direct feedback alignment, and weight mirroring. From this review, it is easy to distinguish the differences between how these signals are combined to generate updates for forward and backward paths.

**Limitations And Societal Impact:**

Modifications to consider based on the main review:

-Adding a formal definition of what constitutes a feedback path.

-A table showing the direct mapping of how the components of the artificial model proposed map to their respective biological counterparts.

-Justification for why identical weight updates is a more reasonable possibility in biological models as compared to identical weights.

-An ablation to show the impact of weight decay on empirical results because it appears to be theoretically necessary (eq 24, eq 25)


**Main Review:**

The algorithm proposed in this work appears to be novel, but the benefit of using this particular idea over others seems to be the avoidance of direct bidirectional connections. Beyond the weight transport problem, this paper suggests that bidirectional connections are also biologically unlikely. But there is no clear definition found in the text of what is strictly a bidirectional connection. From a full read-through, it may be defined as both the feedforward and feedback paths having a direct co-dependency with one another over the same set of variables (h, \delta). But this definition of directionality is rather weak because any chain a 3-step chain of dependency would escape this constraint. The proposed activation sharing algorithm has a 3-step dependency (see Figure 1b h -> \delta_{l+2} -> \delta_{l+1} ). It is unclear why a 3-step dependency is more justifiable than a direct two-point dependency.

What constitutes a feedforward path vs a feedback pathway in biological systems? Given any randomly selected patch of cortex, what would qualify a neuron as a feedforward as opposed to feedback? Is there any identifiability of these classes of pathways that translate into analogous artificial neural network counterparts? Answers to this are critical to currently justify this algorithm over alternatives.

While removing constraints to backpropagation and other algorithms is useful in its own right, an analysis empirical or theoretical of the utility of removing constraints would help justify this algorithm beyond simply being more 'biologically plausible'.

Overall, the Forward Alignment concept proposed in this work is creative and shows another way backpropagation can be modified drastically and still support reasonable efficiency in learning. The work also does well in building upon previous efforts in this direction and adopting aspects like asymmetric paths into the proposed learning method.

**Time Spent Reviewing:**

4

---

> ### Author Response · Authors · 2021-08-10
> **Reference to Reviewer Fjkm**
>
> [1] Mohamed Akrout, Collin Wilson, Peter Humphreys, Timothy Lillicrap, and Douglas B Tweed. Deep learning without weight transport. In Advances in Neural Information Processing Systems, volume 32, 2019.
>
> [2] Andrey Gushchin and Ao Tang. Total wiring length minimization of c. elegans neural network: a constrained optimization approach. PloS one, 2015.
>
> [3] Marion Langen, Egemen Agi, Dylan J Altschuler, Lani F Wu, Steven J Altschuler, and Peter Robin Hiesinger. The developmental rules of neural superposition in drosophila. Cell, 2015.
>
> [4] Sanford L Palay and Victoria Chan-Palay. Cerebellar cortex: cytology and organization. 2012.
>
> [5] Joao Sacramento and Rui Ponte Costa and Yoshua Bengio and Walter Senn. Dendritic cortical microcircuits approximate the backpropagation algorithm. NeurIPS 2018, Montréal, Canada (pp. 8735–8746).
>
> [6] György Buzsáki and Kenji Mizuseki. The log-dynamic brain: how skewed distributions affect network operations. In Nature Reviews Neuroscience, pages 264—-278. 2014
>
> [7] David Fitzpatrick. The Functional Organization of Local Circuits in Visual Cortex: Insights from the Study of Tree Shrew Striate Cortex. Cerebral Cortex, 1996.
>
> [8] Seung Wook Oh, Julie A. Harris, and Hongkui Zeng. A mesoscale connectome of the mouse brain. In Nature, pages 207—-214. 2014.
>
> [9] Alex M. Thomson. Neocortical layer 6, a review. Front. Neuroanat, 2010.
>
> [10] Sato, T. K. Long-range connections enrich cortical computations. Neuroscience Research, 162, 1-12, 2021.
>
> [11] Kathleen S Rockland and Agnes Virga. Terminal arbors of individual “feedback” axons projecting from area v2 to v1 in the macaque monkey: a study using immunohistochemistry of anterogradely transported phaseolus vulgaris-leucoagglutinin. Journal of Comparative 392 Neurology, 1989.
>
> [12] Penelope C Murphy and Adam M Sillito. Functional morphology of the feedback pathway from area 17 of the cat visual cortex to the lateral geniculate nucleus. Journal of Neuroscience, 1996.
>
> [13] Rockland, K. S., Kaas, J. H., & Peters, A. (Eds.). Cerebral Cortex: Volume 12: Extrastriate Cortex in Primates (Vol. 12). Springer Science & Business Media, 2013.
>
> [14] Timothy P Lillicrap, Daniel Cownden, Douglas B Tweed, and Colin J Akerman. Random synap tic feedback weights support error backpropagation for deep learning. Nature communications, 2016.
>
> [15] Arild Nøkland. Direct feedback alignment provides learning in deep neural networks. In  Advances in Neural Information Processing Systems, volume 29, 2016.
>
> [16] Theodore H. Moskovitz, Ashok Litwin-Kumar, and L. F. Abbott. Feedback alignment in deep convolutional networks. CoRR, 2018
>
> [17] Julien Launay, Iacopo Poli, François Boniface, and Florent Krzakala. Direct feedback alignment scales to modern deep learning tasks and architectures. In Advances in Neural Information Processing Systems, volume 33, 2020.
>
> [18] Gilbert, C. D. & Li, W. Top-down influences on visual processing. Nat. Rev. Neurosci. 14, 350–363, 2013.
>
> [19] Tong, F. Primary visual cortex and visual awareness. Nat. Rev. Neurosci. 4, 219–229, 2003.
>
> [20] Lillicrap, T. P., Santoro, A., Marris, L., Akerman, C. J., & Hinton, G. Backpropagation and the brain. Nature Reviews Neuroscience, 21(6), 335-346, 2020
>
> [21] Guillery, R. & Sherman, S. M. Thalamic relay functions and their role in corticocortical communication: generalizations from the visual system. Neuron 33, 163–175, 2002.
>
> [22] Sherman, S. M. & Guillery, R. Distinct functions for direct and transthalamic corticocortical connections. J. Neurophysiol. 106, 1068–1077, 2011.
>
> [23] Viaene, A. N., Petrof, I. & Sherman, S. M. Properties of the thalamic projection from the posterior medial nucleus to primary and secondary somatosensory cortices in the mouse. Proc. Natl Acad. Sci. USA 108, 18156–18161, 2011.
>
> [24] Crick, F. The recent excitement about neural networks. Nature 337, 129–132, 1989.

---

> ### Author Response · Authors · 2021-08-10
> **Response to Reviewer Fjkm**
>
> We thank the reviewer for their time in carefully reviewing our submission and providing valuable comments. Please find our responses below:
> \
> \
> **Q1)** There is no clear definition found in the text of what is strictly a bidirectional connection.
> \
> \
> **A1)** We apologize for the confusion. A bidirectional connection represents a connection between two neurons explicitly exchanging information with each other through a pair of direct unidirectional paths, as shown in Figure 1d.  Due to bidirectional connections, feedforward and feedback paths have a direct co-dependency over the same set of variables, as pointed out by the reviewer. If accepted, we will clarify this point in the final version.
> \
> \
> \
> \
> **Q2)** Bidirectional connections could be realized using a 3-step chain dependency, and the proposed ASAP algorithm also has a 3-step dependency. Then why is ASAP more biologically plausible than the algorithms having bidirectional connections? Why is a 3-step dependency more justifiable than a two-level dependency?
> \
> \
> **A2)** We thank the reviewer for raising a very important issue. We apologize that our original manuscript did not discuss this point in detail. Akrout et al. [1], who proposed algorithms using the bidirectional connections, claim that one-to-one paring that biologically supports a bidirectional connection is observed in some organisms [2, 3, 4]. Although we agree that bidirectional connections may exist in the biological neural network, requiring those connections still impose structural limitations in constructing neural networks, as discussed in other prior works [2, 5]. In fact, Akrout et al. [1] also acknowledge this limitation and mention that “something less than strict one-to-one wiring may suffice for effective learning, and may itself be learned”.
>
> Our ASAP algorithm could significantly relax these structural constraints. ASAP does not require bidirectional connections, and activation sharing can be implemented using a one-way skip connection (i.e., a connection between non-adjacent layers). Those one-way skip connections are frequently found in living organisms [6, 7, 8, 9]. In addition, it has recently been confirmed that long-range connections play an important role in cortical computation [10]. The observations in the visual cortex also show that reciprocal connections are generally spatially asymmetric, consisting of multi-step one-way connections [11, 12].
>
> The reviewer is certainly correct that ASAP also has dependencies between neurons through multiple steps. Nevertheless, we believe that our algorithm is meaningful in that it can relax structural restrictions by showing that it could learn sufficiently without strict two-step dependencies. In fact, multi-step dependencies are often observed in the actual visual cortex. For example, feedforward terminations to layer 4 do not connect directly with the feedback neurons of layer 3A, but only with the feedback neurons of layer 6 along the distal extent of their dendrites [13]. From these observations, Rockland et al. [13] argue that feedforward termination probably uses multiple short- and long-chain and interlaminar routes to connect to feedback neurons for any two interconnected areas. We expect that our ASAP algorithm, which relaxes the structural constraints of strict two-step dependencies, coincides well with such complex biological systems. These points will be later reflected in the camera-ready version.
> \
> \
> \
> \
> **Q3)** What constitutes a feedforward path vs a feedback pathway in biological systems? Is there any identifiability of these classes of pathways that translate into analogous artificial neural network counterparts?
> \
> \
> **A3)** This is also an important issue in biological learning algorithms. Whether a feedback path exists and is identifiable is a challenging question, and many neuroscientists are still trying to find concrete evidence. Since Lillicrap et al. [14] proposed Feedback Alignment that realizes learning through a feedback path, research on implementing neural network architecture with a similar feedback path has continued [15, 16, 17]. Although these algorithms generally have good performance compared to other algorithms that do not assume a feedback path, as the reviewer pointed out, the evidence of the feedback path still remains limited. Of course, in biological neural networks, top-down feedback connections are often found in some areas, such as the visual cortex [18, 19]. However, this observation does not necessarily translate into the fact that a feedback connection actually implements 'error propagation'.
>
> &nbsp;&nbsp;&nbsp;&nbsp;One possible answer is that feedback connections may modulate the spiking of ‘feedforward’ neurons directly, rather than explicitly propagating errors [20]. These feedback connections could be top-down cortico-cortical processing areas, such as those that exist between V2 and V1 in the visual system [18]. The cortico-thalamo-cortical loop might also be a possible alternative, transmitting higher-order information to the cortical region and lower-order information to individual neurons [21, 22, 23]. However, these are still not concrete evidence supporting an algorithm using feedback paths, including our ASAP algorithm, as a biological counterpart. If the reviewer’s question could be solved, it will mark a milestone in the field of neuroscience. Therefore, we will continue research to solve this question. Further, this discussion will be reflected in the updated manuscript to inform the readers of the importance of the existence and identifiability of feedback paths.
> \
> \
> \
> \
> **Q4)** Are identical weight updates more biologically justified than identical weights?
> \
> \
> **A4)** Before answering the question, we need to understand why weight transport is difficult to implement biologically. For backpropagation to work in neural networks, the feedforward and feedback neurons must share the same synaptic weights. This could be realized by explicitly exchanging synaptic weights between the two paths (i.e., weight transport). However, it has been believed that explicitly transporting weight or weight updates is not supported in biological neural networks as it requires a very fast transmission of information along the axon from each synapse output [24]. Also, exact patterns of reciprocal connectivity have not yet been discovered in the brain [24].
>
> &nbsp;&nbsp;On the other hand, Akrout et al. [1] show that there is a way to align weight updates indirectly by transporting forward activation and feedback errors, which are later used for weight update calculation, rather than directly transporting weight updates. Similarly, our ASAP algorithm supports information transport between neurons, but without requiring bidirectional connections. This could alleviate the structural constraints caused by bidirectional connections. This point will be reflected in the manuscript later.
> \
> \
> \
> \
> **Q5)** An ablation to show the impact of weight decay on empirical results is needed.
> \
> \
> **A5)** We thank the reviewer for the helpful suggestion. All the experiments reported in the original paper employed weight decay. Following the reviewer's advice, we additionally performed the same experiments without weight decay to verify its effectiveness. When ResNet-34 is trained on the CIFAR-10 dataset, the experimental results are as follows:
> \
> \
> &nbsp;&nbsp;&nbsp;&nbsp;&nbsp;&nbsp;BP with weight decay: 95.2%
>
> &nbsp;&nbsp;&nbsp;&nbsp;&nbsp;&nbsp;BP without weight decay: 93.56% (-1.14%)
>
> &nbsp;&nbsp;&nbsp;&nbsp;&nbsp;&nbsp;ASAP with weight decay: 93.97%
>
> &nbsp;&nbsp;&nbsp;&nbsp;&nbsp;&nbsp;ASAP without weight decay: 89.77% (-4.2%)
> \
> \
> &nbsp;&nbsp;&nbsp;&nbsp;In both cases, weight decay does improve training performance, but the performance gap is more significant in ASAP. This is consistent with what we expected, because we need a weight decay term to reduce the differences in initial weights during training, as demonstrated in Section 3.3. We plan to conduct more experiments with various weight decay parameters, and they will be reported in the camera-ready version if accepted.

---

### Decision · Program_Chairs · 2021-09-27

**Decision:**

Accept (Poster)

**Comment:**

In this paper the authors present a solution to the problem of bidirectional connections that exist in some solutions to the weight transport problem. More specifically, Akrout et al. (2019) (https://arxiv.org/abs/1904.05391) proposed two algorithmic solutions to the weight transport problem that involved training a separate "error pathway" of neurons, wherein each neuron in the forward processing pathway had a paired error neuron that it was bidirectionally connected to. This is potentially problematic from a biological perspective, as there is no evidence for such a paired error pathway in the neocortex, and moreover, there is no guarantee of bidirectional connections in real neural networks. The present paper solves the bidirectional connection problem (but not the paired error pathway problem) by allowing feedforward neurons to project to multiple error neurons as long as they are equal to or higher in the hierarchy. The authors show that weight alignment and learning can still proceed fairly well under these conditions.

This paper received mixed reviews on the borderline. One of the reviewers was fairly positive, two were borderline accepts, and one felt it was a clear reject. The authors' responses were thorough and clear enough that one of the borderline reviewers updated their score from a 6 to a 7, leading to a final average score of 6 (i.e. a borderline accept). As such, this is a borderline case as an AC. This is especially so because biological plausibility is a tricky concept and sometimes hard to pin down. Generally, it is hard to justify rejecting a paper on biological plausibility alone. And, this paper appears to be technically sound. But, biological plausibility is also not a concept without any meaning, and there are some basic aspects of neurophysiology that we can generally employ when asking about biological plausibility. Moreover, when a paper is framed as having its central contribution be that it is more biologically plausible than previous algorithms the question of biological plausibility necessarily becomes the focus.

Nonetheless, after all these considerations, and after a lot of discussion, an accept decision was reached. Ultimately, the paper is a sound contribution to the field and deserves to be published.

However, given the reviews, the authors may want to consider the following issues for when they prepare the camera ready version though:

1) The paper is focused 100% on solving a problem that exists for the algorithms proposed in the paper of Akrout et al. (2019). Only this paper assumes bidirectional connections between the backward and forward pathway neurons. Other papers on weight alignment (such as: https://www.nature.com/articles/s41593-021-00857-x, https://openreview.net/forum?id=rJxWxxSYvB, http://proceedings.mlr.press/v119/kunin20a.html and https://arxiv.org/abs/2005.04168) don't use a separate error pathway with paired neurons and bidirectional connections. Thus, at face value, this paper's contribution is really just to address a gap in biological plausibility introduced by a specific paper (Akrout et al. 2019). That's fine, but it makes the potential impact much more limited. Can the authors articulate how their approach may be used more broadly than to just solve the bidirectional connection issue in Akrout et al.?

2) The proposed solution is not clearly all that more biologically plausible than that proposed by Akrout et al., based on well-known facts of neurophysiology. Specifically, it appears that this model still mandates that an error pathway with paired neurons for the feedforward pathway exists (i.e. each feedforward neuron has an error neuron partner), and this was arguably the most problematic aspect of the Akrout et al. proposal, since there is zero empirical evidence for such paired pathways in the brain. Moreover, there are other biological plausibility issues that this paper introduces, as pointed out by the critical reviewers, such as the errors now being synapses specific and the need for additional feedforward pathways with the existing (and as noted) biologically implausible paired error pathway neurons. It would be good for the authors to address these concerns more in the final version.

In summary, this paper is technically sound, and potentially interesting, so it is an accept. But, per some of the reviews, at face value it is focused on solving a problem in a specific model, and there are still issues with biological plausibility, so the authors should consider these issues when preparing the camera ready version.